

# Insights into Greenland Ice Sheet Surface Roughness from Ku-/Ka-band Radar Altimetry Surface Echo Strengths

Kirk M. Scanlan[1], Anja Rutishauser[2], Sebastian B. Simonsen[1]

[1]DTU Space, Technical University of Denmark, Kgs. Lyngby, 2800 Denmark
[2]Geological Survey of Denmark and Greenland, Copenhagen, 1350, Denmark

*Correspondence to*: Kirk M. Scanlan (kimis@dtu.dk)

**Abstract.**

Surface roughness is an important factor to consider when modelling mass fluxes at the Greenland Ice Sheet (GrIS) surface (i.e., surface mass balance, SMB). This is because it can have important implications for both sensible and latent heat fluxes
between the atmosphere and the ice sheet and near-surface ventilation. While surface roughness can be quantified from ground-based, airborne and spaceborne observations, satellite radar datasets provide the unique combination of long-term, repeat observations across the entire GrIS and insensitivity to illumination conditions and cloud cover. In this study, we investigate the reliability and interpretation of a new type of surface roughness estimate derived from the analysis of Ku- and Ka-band airborne and spaceborne radar altimetry surface echo powers by comparing them to contemporaneous laser altimetry
measurements. Airborne data are those acquired during the 2017 and 2019 CryoVEx campaigns while the satellite data (ESA CryoSat-2, CNES/ISRO SARAL, and NASA ICESat-2) are those acquired in November 2018. Our results show that because surface roughness across the GrIS is primarily scale-dependent, a revised empirical mapping of quantified radar backscattering to surface roughness gives a better match to the coincident laser altimetry observations than an analytical model that assumes scale-independent roughness. We also show that the radar altimetry-derived surface roughness is best interpreted as the
wavelength-baseline linear projection of the scale-dependent surface roughness observed at hundreds of meter scales and is therefore not representative of individual small-scale features. These results provide critical context for interpreting the datasets and evaluating their applicability in modelling GrIS SMB.

## 1 Introduction

The satellite mass balance history of the Greenland Ice Sheet (GrIS) is a record of the balance between loss of snow, firn and
ice due to runoff, evaporation, sublimation, erosion, and calving, and gain in the way of new precipitation (Otosaka et al., 2023; The IMBIE Team, 2020). It is the primary means of understanding Greenland's recent (e.g., 1990's-present) contribution to global sea-level rise and forms the basis for understanding ice sheet evolution in the future (Bamber et al., 2019; Edwards et al., 2021; Goelzer et al., 2020). There are three geodetic ice sheet mass balance observations: gravimetry, input-output, and altimetry (Otosaka et al., 2023). Gravimetric mass balance is based on monitoring changes in the gravity field across the ice



sheet as mass lost through melting/discharge and new snow results in minute but detectable gravity fluctuations. The input-output method compares the quantity of material added or lost to an ice sheet's surface (surface mass balance, SMB) to the amount of ice flowing through peripheral fluxgates. Finally, the altimetric mass balance approach is based on converting changes in ice sheet surface elevation (i.e., volume changes) to changes in mass. Critically, both the input-output and altimetry-based mass balance approaches require model-based estimates of the ice sheet SMB, which in the former defines the net mass

change across the ice sheet surface and in the latter, the non-mass change component of the observed surface elevation change (i.e., due to firn compaction). Regardless, to quantify GrIS mass balance using either the input-output or altimetry approaches, insight into pan-ice sheet SMB is paramount and is typically derived using numerical regional climate models (RCMs) (Alexander et al., 2019; Boberg et al., 2022; Bougamont et al., 2005; Ettema et al., 2009; Medley et al., 2022; Vernon et al., 2013).

Numerical estimates of GrIS SMB represent the intersection of climate and subsurface models; the climate model generates the forcing (e.g., temperature, precipitation, etc.) applied to the ice sheet's surface (Van Den Broeke et al., 2023), while the subsurface model propagates it into the near-surface conditions. In this framework, a critical parameter to constrain is the roughness of the ice sheet surface as it modulates the energy balance between the atmosphere and the subsurface via sensible and latent heat fluxes along with near-surface ventilation (i.e., interstitial airflow through snow/firn) (Albert and Hawley, 2002;

Amory et al., 2016; Braithwaite, 1995; Jakobs et al., 2019; Smeets and Van Den Broeke, 2008; van Tiggelen et al., 2021; Van Der Veen et al., 2009; Van Tiggelen et al., 2023). Outside of affecting atmosphere-ice sheet heat fluxes, surface roughness can also denote different morphogenetic areas of the ice sheet (Nolin et al., 2002; Van Der Veen et al., 2009), steer incipient supraglacial meltwater flow in the ablation zone (Cathles et al., 2011), and affect conventional laser and radar altimetry measurements of surface elevation change (Herzfeld et al., 2000; Van Der Veen et al., 2009; Yi et al., 2005).

Compared to other SMB-relevant parameters (e.g., temperature, precipitation, short-/longwave radiation, wind speed, etc.) surface roughness is unique because 1) it can be dependent on the scale over which it is quantified and 2) there are a variety of different metrics that have been used (Shepard et al., 2001). The most direct means of measuring GrIS surface roughness is with extremely local (i.e., meter-long) comb gauges/snow blades that produce centimetre-scale, one-dimensional replicas of the snow surface (Albert and Hawley, 2002; Jezek, 2007). Relevant statistical descriptions for surface roughness (e.g., peak-

to-peak amplitude, roughness wavelengths, etc.) can then be quantified from these replicas. To extend the meter-scale local comb gauge/snow blade measurements, Herzfeld et al. (2000) developed a towed sensor capable of measuring surface elevations at fine spatial scales along profiles hundreds of meters in length. While ground-based surface roughness measurements yield the finest spatial sampling, their large-scale applicability is limited as they are very time-consuming and subject to site accessibility (e.g., remoteness, weather, etc.) issues.

Furthermore, acquiring data on regional or pan-GrIS scales is extremely relevant for SMB flux calculations (Van Den Broeke et al., 2023). For this, remote sensing via airborne or satellite methods is the sole viable option. Photogrammetry and laser scanning from the air (i.e., drones, helicopters, planes) have each been used in regional surface roughness studies (Nolin et al., 2002; van Tiggelen et al., 2021; Van Der Veen et al., 2009). Still, similar to the in-situ methods, accessibility throughout the



year remains a challenge due to weather and illumination conditions. Satellite remote sensing can acquire data across the GrIS throughout the year and, while optical and laser methods are still sensitive to illumination conditions (e.g., during the polar night) and cloud cover, respectively, radar techniques can operate year-round. GrIS surface roughness has been derived from various datasets collected by different satellites, including US Navy Geosat (Davis and Zwally, 1993), NASA ICESat (Yi et al., 2005), NASA ICESat-2 (van Tiggelen et al., 2021), and NASA Terra (Nolin et al., 2002). It is important to consider though that while satellite remote sensing datasets can be used to characterise surface roughness over large areas, the horizontal sampling of the roughness is typically much coarser than ground-based methods. For example, the horizontal scales over which roughness is measured can vary from 1 m (van Tiggelen et al., 2021) to 10 km (Yi et al., 2005).

Recently, Scanlan et al. (2023) presented a new approach for characterising the monthly variability in surface roughness across the GrIS via the strength of radar altimetry surface echoes. Their approach is based on the Radar Statistical Reconnaissance (RSR) method (Grima et al., 2012, 2014a, 2022). As the name implies, RSR is a statistical approach that allows for the observed strength of radar echoes to be decomposed into their coherent and incoherent components and, when combined with a backscattering model, be used to derive the relative dielectric permittivity and RMS height of the surface. Initially developed to study Mars (Grima et al., 2012, 2022), the RSR method has also been applied to airborne VHF measurements of polar ice masses (Chan et al., 2023; Grima et al., 2014a, b, 2016, 2019; Rutishauser et al., 2016) and Ku-band radar altimetry measurements of the surface of Titan (Grima et al., 2017). Where Scanlan et al. (2023) outline how the RSR method has been implemented for the analysis of Ku-/Ka-band radar altimetry measurements and performs a preliminary qualitative interpretation, this study focuses more intensely on the GrIS surface roughness results, with the specific goal of validating their derivation and interpretation. Only once the foundation of the surface roughness results has been solidified, can their applicability with respect to SMB modelling be explored.

In this study, the reliability of the Ku- and Ka-band surface roughness estimates derived from the RSR analysis of radar altimetry surface echoes is assessed by comparing them to conventional surface roughness metrics derived from laser altimetry measurements. This comparison is performed at both 1) locations of simultaneous airborne radar and laser altimetry collected as part of the 2017 and 2019 ESA CryoVEx campaigns in central Greenland as well as 2) at a set of locations spanning the Greenland ice sheet using satellite (i.e., ESA CryoSat-2, CNES/ISRO SARAL, and NASA ICESat-2) datasets acquired in November 2018. The satellite comparison focuses on November 2018 as the RSR results are generated monthly (Scanlan et al., 2023) and November 2018 is one of the first full months where ICESat-2 was completely operational following its launch two months earlier. We perform the comparison of radar and laser altimetry-derived surface roughness for both the airborne and spaceborne cases for two reasons: first, because the spatial resolution of the airborne datasets is much finer than the spaceborne datasets and second because the spatial coverage of the satellite data is much broader than that of the airborne datasets. Therefore, considering both yields a more complete understanding of the radar altimetry-derived surface roughness results.



## 2. Datasets

### 2.1 CryoVEx Altimetry

The series of CryoVEx (CryoSat Validation Experiment) campaigns operated by ESA focus on 1) validating satellite altimetry measurements over ice sheets and sea ice by way of repeated satellite ground track under flights and 2) supporting the design

of future satellite altimetry missions (e.g., ESA CRISTAL, Sentinel-3 Next Generation Topography (S3NG-T)). To this end, through its history, the CryoVEx nadir-pointed airborne altimetry sensor package has included versions of different altimeters, but the most important for this study (Table 1) are ESA's Ku-band (13.5 GHz centre frequency, 0.1098 m range resolution) ASIRAS radar, the MetaSensing Ka-band (34.525 GHz centre frequency, 0.165 m vertical resolution) KAREN radar, and the Riegl LMS Q-240i-60 Airborne Laser Scanner (ALS; 904 nm wavelength). All altimetry measurements are supported by

precise aircraft positioning by combined GPS and INS (inertial navigation system) navigation solutions. In addition to the airborne platform, CryoVEx campaigns also include a substantial ground component tasked with installing corner reflectors on the surface and taking in-situ measurements (e.g., density) in the shallow subsurface along the aircraft/satellite ground track. This study uses March/April 2017 (Skourup et al., 2019) and April 2019 (Skourup et al., 2021) CryoVEx data (ESA, 2022a, b) collected along the EGIG line in central Greenland. Specifically, it focuses on ASIRAS, KAREN, and ALS measurements

surrounding positions of the in-situ measurements performed at the T5, T9, T12, T19, T30, and T41 locations in 2017 as well as T9, T12, T21, and T35 locations in 2019. Both the ASIRAS and KAREN data have been subject to post-acquisition SAR (synthetic aperture radar) processing to minimise the size of their respective footprints in the alongtrack (3 m and 5 m, respectively) and crosstrack (10 m and 12 m, respectively) directions. The ALS data have been processed into 200-300 m wide swaths following the aircraft ground track, with individual ground height measurements spaced at one-meter by one-meter

intervals.

### 2.2 Satellite Altimetry

For decades, satellite radar and laser altimetry have been the method-of-choice for acquiring the ice sheet surface elevation change (SEC) measurements that feed long-term mass balance monitoring efforts (Abdalati et al., 2010; Markus et al., 2017; Otosaka et al., 2023; Schröder et al., 2019; Schutz et al., 2005; The International Altimetry Team, 2021). Of particular interest

in this study (Table 1) are the Ku-band radar altimetry datasets from the SIRAL instrument (13.575 GHz centre frequency, 320 MHz bandwidth) on-board the ESA CryoSat-2 spacecraft (Phalippou et al., 2001; Rey et al., 2001; Wingham et al., 2006) as well as the Ka-band measurements from the AltiKa instrument (35.75 GHz centre frequency, 500 MHz bandwidth) on-board the CNES/ISRO SARAL spacecraft (Steunou et al., 2015; Verron et al., 2015). Laser altimetry measurements come from the ATLAS instrument (532 nm wavelength) on-board the NASA ICESat-2 spacecraft (Abdalati et al., 2010; Markus et

al., 2017).

Leveraging the results of Scanlan et al. (2023), because CryoSat-2 operates in two different acquisition modes while over Greenland, this study makes use of both Low-Resolution Mode (LRM) Level 1B (L1B) and SAR Interferometric (SARIn)



Full-Bit Rate (FBR) CryoSat-2 Baseline-D data products. The LRM data products cover the GrIS interior, where CryoSat-2 waveforms are stacked from an individual 1.97 kHz pulse repetition frequency (PRF) to a constant data rate of 20 Hz. Across

the GrIS margins, the CryoSat-2 SARIn data products contain reflected waveforms from 18.181 kHz PRF, 64-pulse bursts (burst repetition frequency of 21 Hz) acquired by both CryoSat-2 receive antennas. The FBR data are preferred to higher level SARIn data products (e.g., Level 1) as they have not yet been subject to any SAR focusing. The SARAL Sensor Geophysical Data Record (SGDR) data products are similar to CryoSat-2 LRM products in that initially 3.8 kHz PRF waveforms are averaged alongtrack in 25 ms (40 Hz) intervals. Finally, this study makes use of ATL06 ICESat-2 data products providing

geo-located surface heights across the GrIS in 20-meter increments.

**Table 1:** Summary of the airborne and satellite radar and laser altimetry datasets used to derive surface roughness as part of this study.

| | Instrument | Altimeter Type | Radar Centre Frequency | Wavelength | Derivation of Surface Roughness | Maximum RSR Search Radius |
|---|---|---|---|---|---|---|
| *Airborne* | | | | | | |
| | CryoVEx ASIRAS | Radar | 13.5 GHz | 2.22 cm | RSR | n/a |
| | CryoVEx KAREN | Radar | 34.525 GHz | 0.86 cm | RSR | n/a |
| | CryoVEx ALS | Laser | n/a | 904 nm | Surface heights | n/a |
| *Satellite* | | | | | | |
| | ESA CryoSat-2 SIRAL | Radar | 13.575 GHz | 2.21 cm | RSR | LRM: 50 km SARIn: 25 km |
| | CNES/ISRO SARAL AltiKa | Radar | 35.75 GHz | 0.84 cm | RSR | 40 km |
| | NASA ICESat-2 ATLAS | Laser | n/a | 532 nm | Surface heights | n/a |

**3. Methods**

**3.1 Quantification of Surface Roughness from Laser Altimetry**

As alluded to previously, there is no best, single method for measuring surface roughness as it can be both scale-dependent as well as scale-independent and multiple metrics have been put forward in the literature. Shepard et al. (2001) provide a thorough overview of various surface roughness metrics and how they can relate to one another. With the horizontal scale dependence

of surface roughness a distinct possibility, this study adopts the strategy of Steinbrügge et al. (2020) and quantifies surface roughness by way of the RMS deviation.





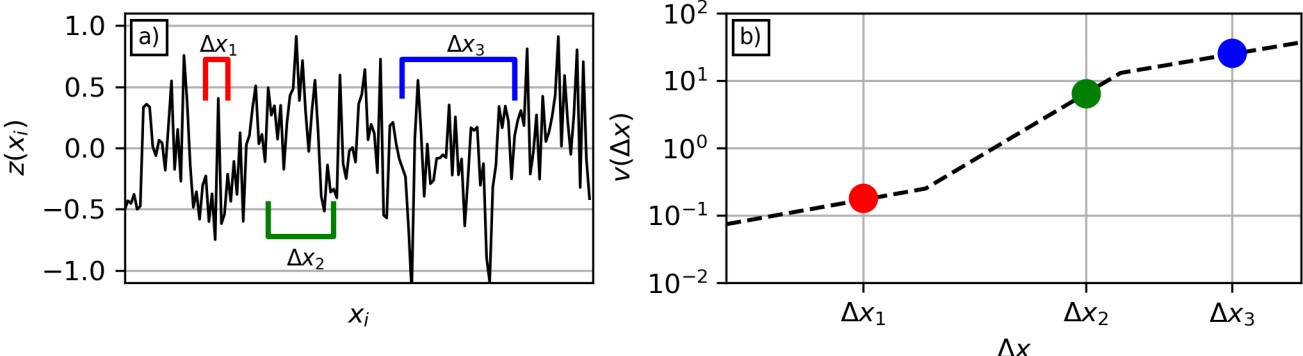

**Figure 1**: Simplified diagram demonstrating how to derive an RMS deviation $[v(\Delta x)]$ profile that quantifies surface roughness as a function of horizontal baseline ($\Delta x$). The profile comprises the RMS of height deviations [i.e., elevations minus a background plane; $z(x_i)$] considering all combinations of points with a similar horizontal baseline. The RMS deviation profile [b)] can be derived from profile data (i.e., as demonstrated here) as well as two-dimensional surface elevation datasets.

Figure 1 presents a simplified overview of how to calculate the RMS deviation at various baselines from a set of measured surface elevations along a profile. First, a constant background plane is removed from the measured surface elevations, yielding a profile of surface deviations [$z(x_i)$ in Figure 1a]. Then the differences in height deviation are calculated for the $n$ possible combinations of points along the profile that are spaced some defined horizontal distance (i.e., baseline) apart (e.g., $\Delta x_1$, $\Delta x_2$, and $\Delta x_3$ in Figure 1). The RMS deviation for that specific baseline is then the RMS of all those height deviation differences following

$$v(\Delta x) = \left\{ \frac{1}{n} \sum_{i=1}^{n} [z(x_i) - z(x_i + \Delta x)]^2 \right\}^{1/2}. \tag{1}$$

Finally, the RMS deviation profile (Figure 1b) presents how the RMS deviations vary as a function of the range of horizontal baselines considered (i.e., the horizontal scale dependency in surface roughness). While Figure 1 presents the derivation of the RMS profile from a one-dimensional profile of surface elevations, the procedure can also be expanded to two-dimensional surface elevation datasets. In such an application, it is possible to derived both isotropic (i.e., baselines in all directions considered equally) and anisotropic (i.e., baselines restricted to only certain azimuth/cardinal directions) RMS deviation profiles. When presented in log-log space, it is common for the RMS deviation profiles (Figure 1b) to exhibit a piecewise linear behaviour (Steinbrügge et al., 2020).

**3.2 Radar Statistical Reconnaissance (RSR) Implementation and Calibration**

As RSR is based on the statistical distribution of measured surface echo powers, the first step in implementation is to extract those surface echo powers from the measured waveforms. The same surface detection and extraction approach is applied to the CryoVEx (ASIRAS and KAREN) and satellite (CryoSat-2 and SARAL) datasets and follows Scanlan et al. (2023). The



leading edge is defined as the point of maximum, integrated rate of change in the waveform amplitude when derived over different proportions of the receive window. For the airborne data, individual rates of change in the receive waveform amplitude are calculated over two and four percent of the receive window length, while for the satellite data, amplitude derivatives over three, six and nine percent of the waveform receive windows are used. The extracted surface echo power is

then the maximum power observed within the five percent of the range window following the re-tracked leading edge position. One feature relevant to the analysis of the airborne ASIRAS data is that the amplitudes provided in the CryoVEx data products have been normalized. Representative waveform amplitudes ($A_{rep}$) are generated from the normalized amplitudes ($A_{norm}$) following

$$A_{rep} = A_{norm} * 10^{-9} * FAC_A * 2^{FAC_B} \, , \qquad (2)$$

where $FAC_A$ and $FAC_B$ are the Linear Scale Factor A and Power of 2 Scale Factor variables reported in the ASIRAS data products. Once extracted, ASIRAS and KAREN surface power amplitudes are omitted from further analysis if the measured roll is greater than 1.5 degrees to limit the potential effects of off-nadir instrument pointing. For the satellite data, CryoSat-2 SARIn echo powers are removed if there is a marked CAL4 flag, while for SARAL, the altimetric range (>1000 km), a trailing edge variation flag, and a large average waveform off-nadir angle (>0.10 degrees-squared) are all used to remove possibly

erroneous surface echo powers. Finally, the satellite surface echo powers are also corrected for the nadir surface slope (Scanlan et al., 2023) using the 500 m ArcticDEM (Porter et al., 2018), while the more local CryoVEx data are not.

Fundamentally, how the RSR method is implemented is the same for all the radar altimetry datasets. The set of radar surface echo amplitudes closest to some defined location is used to construct a density function and the statistical descriptors of the homodyned K-distribution fit define the coherent ($P_c$) and incoherent ($P_n$) powers (Grima et al., 2014a). For the CryoVEx case,

the locations are defined as the positions of in-situ density measurements along the EGIG line (Skourup et al., 2019, 2021). The choice to focus exclusively on the immediate area surrounding in-situ measurements allows for the rapid check of the reliability of the RSR approach when applied to CryoVEx data by way of reproducing in-situ density estimates. For both ASIRAS and KAREN, $P_c$ and $P_n$ are derived from the distribution of the 1000 closest surface echo powers to each in-situ position (measured in EPSG:3413). For the satellite radar altimetry datasets (CryoSat-2 LRM, CryoSat-2 SARIn, SARAL),

this study makes partial use of the results from Scanlan et al. (2023), where RSR has been applied to the 1000 closest surface echo powers surrounding nodes of a five-by-five kilometre grid spanning the GrIS. For the CryoSat-2 SARIn data, the same five-by-five kilometre set of grid nodes are used, but instead of 1000 samples, RSR results are derived from the 12,000 closest samples to overcome the apparent statistical dependence of adjacent CryoSat-2 SARIn surface echo powers noted in Scanlan et al. (2023).

Two metrics are used to ensure the quality of the RSR results: first, the maximum distance one must search around the defined location to collect the required number of surface echo powers, and second, the correlation coefficient between the observed surface echo power histogram and the homodyned K-distribution fit. The former is irrelevant for the CryoVEx data as we mandate using the closest 1000 data points surrounding the in-situ locations regardless of how far they may be. For the satellite results, we use the following maximum search radii: 50 km for CryoSat-2 LRM, 25 km from CryoSat-2 SARIn, and 40 km for





SARAL (Table 1). The CryoSat-2 LRM and SARAL maximum search radii are the same as those used in Scanlan et al., (2023), while the CryoSat-2 SARIn radius has been extended to account for the increased number of echo samples considered. For all radar altimetry datasets, the correlation coefficient must exceed 0.96 for the RSR measurement to be considered valid (Grima et al., 2012, 2014b, a; Scanlan et al., 2023).

From the quality controlled RSR results, surface roughness is derived by adopting a representative backscattering model. The

simplest implementation of the RSR technique (Grima et al., 2012, 2014b, a; Scanlan et al., 2023) assumes incoherent backscattering from the surface follows the Small Perturbation Model (SPM) (Ulaby et al., 1982) where the surface RMS height can be derived from the ratio of the coherent and incoherent powers following

$$\sigma_h = \frac{\lambda e^{P_n/2P_c}}{4\pi\sqrt{P_c/P_n}},$$                                                  (3)

where $\lambda$ is the signal wavelength [m]. The validity bounds of the SPM are $k\sigma_h < 0.3$ and $kl < 3$ where $k$ is the radar

wavenumber [m$^{-1}$] and $l$ is the surface roughness correlation length [m]. The analytical relationship for deriving RMS height from the RSR results in Equation 3 is based on the assumption that the surface roughness correlation length is large enough for its influence to be neglected (Grima et al., 2012, 2014a). Note that in contrast to when deriving surface dielectric permittivities from the RSR results, the absolute calibration of the RSR coherent powers is not directly required to produce a roughness estimate (Grima et al., 2012, 2014a, b; Scanlan et al., 2023). However, as an additional check on the overall

applicability of the RSR approach, CryoVEx RSR results have been calibrated using the contemporaneous in-situ measurements. The CryoVEx $P_c$ and associated calibration density values vary between 2017 (189.5 dB and 0.438 g cm$^{-3}$ for ASIRAS; 324.6 dB and 0.353 g cm$^{-3}$ for KAREN) and 2019 (183.7 dB and 0.57 g cm$^{-3}$ for ASIRAS; 317.25 dB and 0.5 g cm$^{-3}$ for KAREN) and are based on the same assumed Ku- and Ka-band density depth sensitivity as used in Scanlan et al. (2023).

**4. Comparisons of Radar and Laser Altimetry Surface Roughness Estimates**

**4.1 Airborne CryoVEx Datasets**

The results of the CryoVEx RSR analysis are presented in Figure 2. All EGIG sites visited in 2017 and 2019 have been analysed. However, it should be noted that in the case of T35 (grey circle in Figure 2a) , which was visited in 2019, the ASIRAS and KAREN RSR results both failed the quality control assessment and no data from this location are presented. The comparison of the measured in-situ average densities (top two meters for KAREN and top four meters for ASIRAS; Scanlan

et al., (2023)) with those derived from the RSR analysis demonstrates we can reasonably recover the in-situ densities from the remote sensing measurements after calibration (Figure 2b). Comparisons of the 2017 and 2019 surface roughness results from the ASIRAS, KAREN and ALS altimetry datasets are presented in Figures 2c and 2d, respectively. It is assumed that the ASIRAS (triangles) and KAREN (squares) RMS heights derived from Equation 3 are equivalent to the ALS RMS deviations at a horizontal baseline equal to the respective radar wavelength (Table 1).




**Figure 2:** Results from the analysis of 2017 and 2019 airborne CryoVEx data. The locations of the in-situ measurements along the EGIG line are presented in Panel a). Panel b) presents the agreement (R of 0.45) between the RSR-derived densities from ASIRAS (triangles) and KAREN (square) surface echoes powers with those measured in-situ. The comparison of RMS heights from the radar altimetry (assumed to be equivalent to the RMS deviation at the wavelength baseline) and the ALS data (circles)

RMS deviations as a function of baseline are presented in Panels c) (CryoVEx 2017) and d) (CryoVEx 2019). The RSR-based roughness estimates align well with the projection of the piecewise linear portion of the ALS RMS deviations profiles between 200-700 m (i.e., the grey region) to the radar wavelength baseline.

It is immediately clear from Figure 2 that the ASIRAS and KAREN surface roughness estimates do not agree with a direct

continuation of the shortest baseline scale laser surface roughness behaviours down to the wavelength scale. While RMS deviation typically exhibits a strong scale dependency for baselines greater than 100 m (i.e., RMS deviation increases with baseline), for shorter baselines, the RMS deviation profiles level off, revealing more scale-independent behaviour. The continuation of the piecewise linear trends in RMS deviation from baselines less than 50 m to the radar wavelengths would



overestimate the RSR results by roughly two orders of magnitude. However, projecting that more scale-dependent behaviour

at larger baselines (e.g., between 200 and 700 m) yields the dashed lines in Figures 2b and 2c, which closely intersect with the RMS heights derived from the ASIRAS and KAREN. **This suggests that the RMS heights derived through the RSR analysis of CryoVEx radar altimetry surface echo powers should be interpreted as the wavelength-scale projection of surface roughness behaviour observed at long baselines and not the RMS deviation at the wavelength scale**. It should be noted, however, that this interpretation assumes that there are no further inflexion points at baselines smaller than those that

can be accessed via the ALS data.

**Figure 3:** Surface elevations [a) and b)] and height deviations [c) and d)] surrounding the T30 [a) and c)] and T41 [b) and d)] CryoVEx locations. The ALS data considered in deriving the corresponding RMS deviation profiles in Figure 2c come from within the black polygons. It is clear that T30 and T41 are sited in locally smooth regions of the GrIS.




Amongst the ALS results, the 2017 T30 and T41 RMS deviation profiles (Figure 2c) are unique, in that they do not exhibit the increased scale-dependent roughness behaviour at longer baselines. Instead, their RMS deviation profiles are flat and monotonic (i.e., not piecewise linear). The reason for this change in surface roughness behaviour is due to the ALS data surrounding T30 and T41 preferentially covering extremely smooth local areas of the GrIS. Figure 3 presents the local surface

elevations (Figures 3a and 3b) along with the height deviations (elevations minus the constant location-specific background plane; Figures 3c and 3d) surrounding the T30 and T41 ALS datasets. Elevations are taken from the 10 m ArcticDEM mosaic (Porter et al., 2018) and the background plane is defined using all 10 m ArcticDEM data within 20 km of the CryoVEx in-situ measurement location. It is clear that the topographic variability across each of these sites is very small at T30 and essentially non-existent at T41. It is then not surprising that the corresponding RMS deviation profiles (Figure 2c) do not exhibit the

increase in RMS deviation at large horizontal baselines that is observed at the other CryoVEx locations. To further emphasise the smoothness of the GrIS near T30 and T41, Figure 4 compares the ALS RMS deviation profiles with those derived from all ICESat-2 surface elevations within 25 km and 35 km of T30 and T41, respectively. We must use ICESat-2 data that are further away from the T30 and T41 because these locations are between ICESat-2 orbital ground tracks. When considering the surface topography over a broader area (i.e., ICESat-2), the increase in surface roughness at longer baselines is once again observed.

It is also worth noting that the breakpoint in the RMS deviation profiles at baselines between 100 and 200 m is not being captured by the 20 m posting ICESat-2 ATL06 data. Therefore, without the small baseline insight gained from the airborne CryoVEx ALS data, the RSR surface roughness results could have been misinterpreted as a direct continuation of the monotonic ICESat-2 RMS deviation profiles.





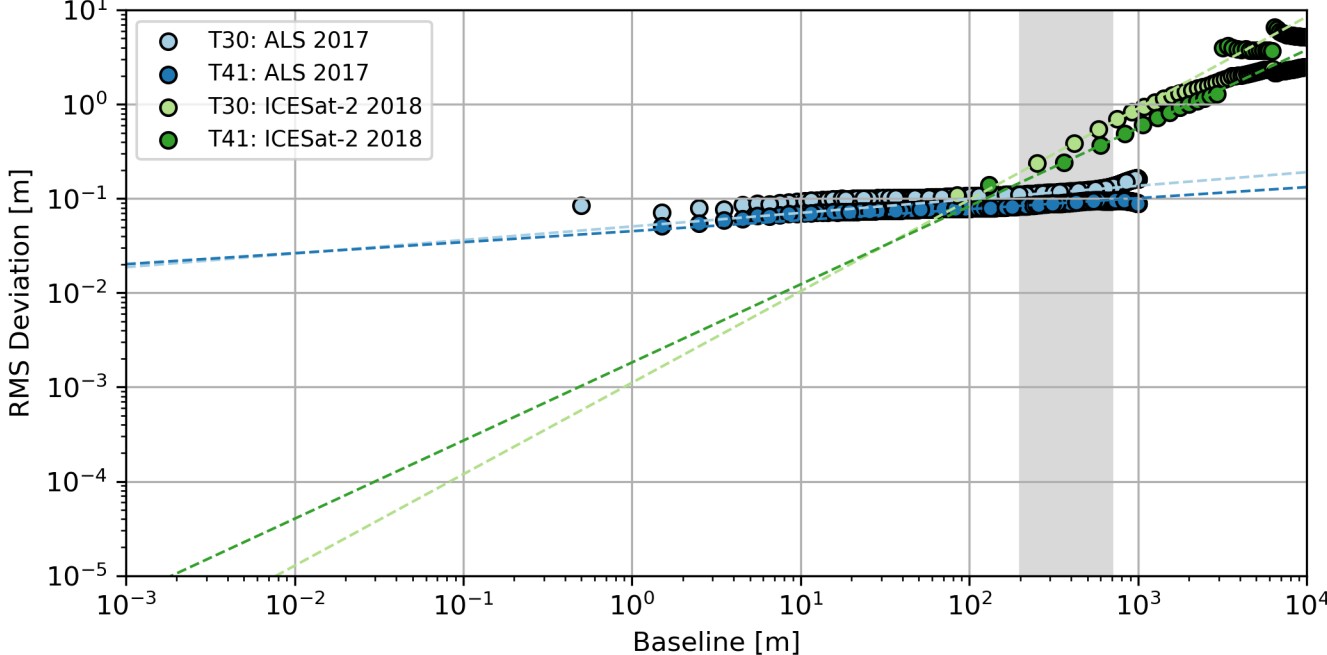


**Figure 4:** The comparison of March/April 2017 ALS (blues) and November 2018 ICESat-2 (greens) RMS deviation profiles centred on locations T30 (light) and T41 (dark) along the EGIG line. There is a substantial discrepancy between the two sets of RMS deviation profiles in the overlapping baseline range (100 m to 1 km) further confirming that the local regions surrounding T30 and T41 are markedly smoother than those further afield. The anomalous behaviour in the ICESat-2 RMS

deviation profiles at large (i.e., >4 km) baselines is related to the quick drop-off in the number of comparable surface elevations.

## 4.2 Spaceborne Datasets

Turning to the spaceborne altimetry datasets (i.e., ESA CryoSat-2, CNES/ISRO SARAL, and NASA ICESat-2), the first step is to derive RMS deviation profiles from the laser altimetry surface elevations. To maintain equivalence in the spatial representativeness of the radar and laser altimetry surface roughness metrics for a specific location in the five-by-five kilometre

grid, ICESat-2 RMS deviation profiles are derived from all November 2018 surface elevations within the three different RSR surface echo power search radii for that location (maximum search radii are presented in Table 1). The number of ICESat-2 measurements within the location-specific CryoSat-2 LRM, CryoSat-2 SARIn and SARAL search radii for November 2018 are presented in Figure 5. As expected, because the spatial density of surface echo power measurements along an individual orbit (and therefore across a month) is greatest for CryoSat-2 when operating in its SARIn mode, the associated smaller search

radii contain the fewest ICESat-2 measurements (Figure 5b). In contrast, the lower spatial resolution (i.e., less frequent alongtrack sampling) inherent in the CryoSat-2 LRM (Figure 5a) and SARAL (Figure 5c) data requires a greater RSR search radius that, in turn, encompasses more ICESat-2 measurements. The cross-cutting striped patterns in Figure 5 relate to the specific distribution of satellite ground tracks across the GrIS in November 2018. As three different sets of ICESat-2 data are



used to derived three RMS deviation profiles for each location, three different background planes are also defined based on
the local ATL06 surface elevations.

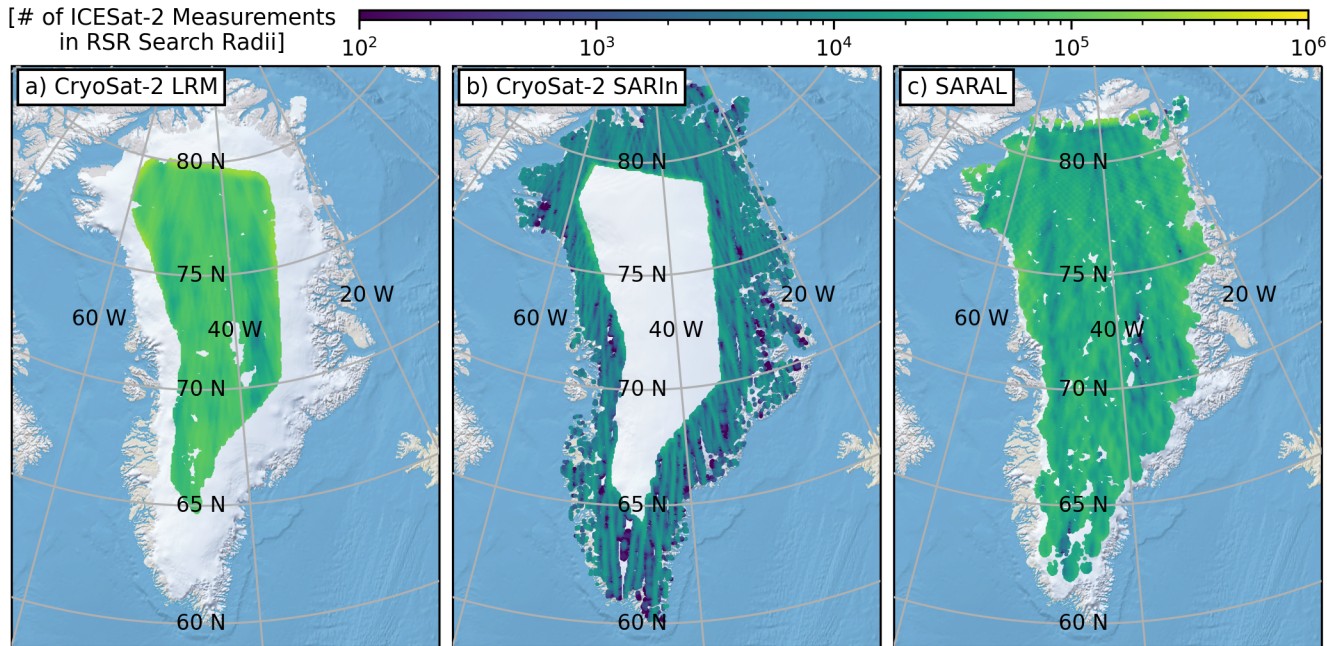

**Figure 5:** Maps showing the number of November 2018 ICESat-2 ATL06 measurements within the contemporaneous a)
CryoSat-2 LRM, b) CryoSat-2 SARIn, and c) SARAL RSR search radii. The quantity of ICESat-2 ATL06 surface elevations
used to derive the RMS deviation profile for a specific location is inversely related to the alongtrack data rates of the different
radar altimeters.

In lieu of presenting hundreds of individual ICESat-2 RMS deviation profiles together with the CryoSat-2 and SARAL RMS
heights (i.e., akin to Figures 2c and 2d) and following on from what has been learned from the CryoVEx results, Figure 6
presents the comparisons of the RSR RMS heights, and the wavelength-scale RMS deviations projected from a linear fit to the
RMS deviation profiles for baselines between 200 and 700 m for 328 locations across the GrIS. These 328 locations have been
pseudo-randomly selected based solely on considerations for the computational load when performing the point-to-point
surface deviation comparison as part of the RMS deviation profile calculation (i.e., ≤55,000 ICESat-2 surface elevations). To
ease the comparison, all surface roughness estimates (i.e., CryoSat-2 and SARAL RSR RMS heights or ICESat-2 RMS
deviations projected to the CryoSat-2 and SARAL wavelength scale) have been normalized by the radar signal wavelength.
While there may be the suggestion of possible linear relationship between the radar- and laser-derived surface roughness
estimates, the is clearly no 1:1 agreement. This appears to be in part due to a floor in the RSR results as they consistently fail



to recover the smallest ICESat-2 RMS deviations. The mean absolute error between the two sets of surface roughness estimates is 0.0308 $\lambda$ for CryoSat-2 and 0.0346 $\lambda$ for SARAL.

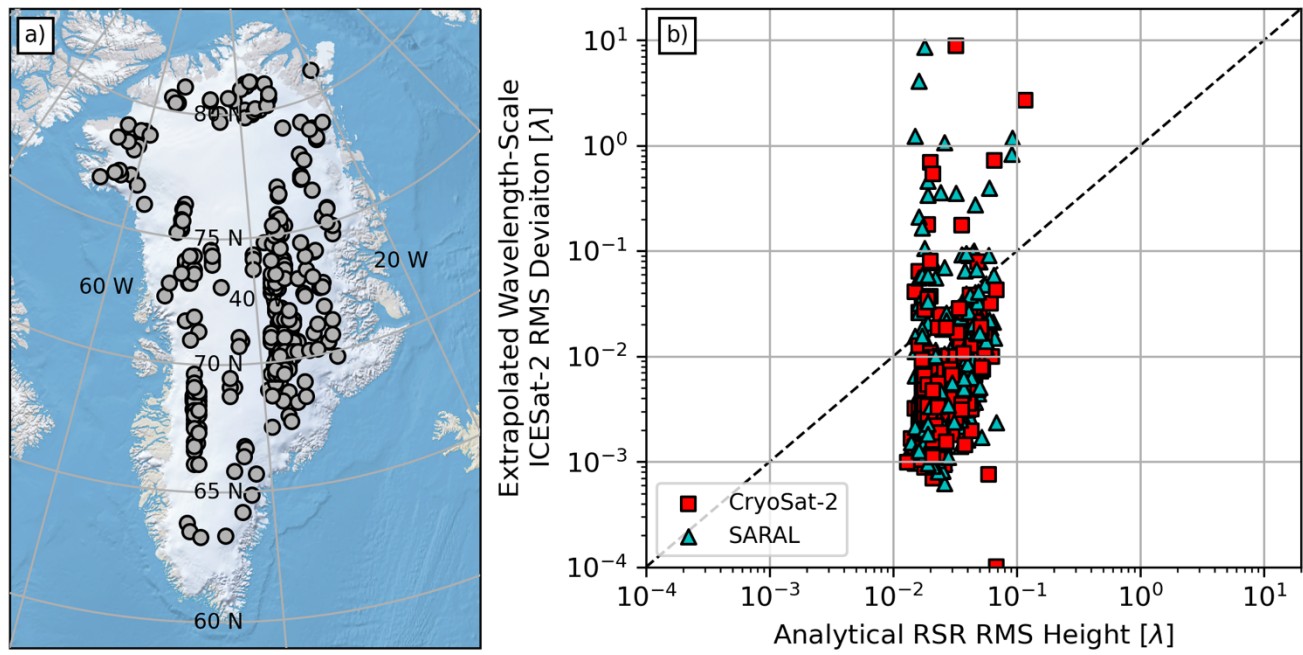


**Figure 6:** Panel a) presents the locations of 328 pseudo-randomly chosen locations across the GrIS where in Panel b) RSR surface roughness results (CryoSat-2 as squares and SARAL as triangles) are compared to wavelength-baseline projected RMS deviations from ICESat-2. While there is a general positive associated between the two sets of roughness estimates, the RSR results do not reliably recover the smallest ICESat-2 roughness levels. The mean absolute error between the ICESat-2 and the

CryoSat-2 and SARAL RSR-based wavelength-normalized roughness estimates are 0.0308 $\lambda$ and 0.0346 $\lambda$ respectively.

## 5. Revising the Derivation of Surface Roughness

### 5.1 An Empirical Roughness Relationship

The clear absence of a good agreement in Figure 6 between the CryoSat-2 (LRM and SARIn) and SARAL RSR results with, what are taken to be, equivalent results derived from ICESat-2 necessitates a deeper investigation into the rationale for why.

As such, building off the general relationship established by Equation 3, Figure 7 presents a two-dimensional histogram directly comparing the CryoSat-2 and SARAL coherent/incoherent power ratios (in linear units plotted on a logarithmic axis) with the radar wavelength-scale RMS deviation projected from the ICESat-2 RMS deviation profile between 200 and 700 m baselines. Following Figure 6, the projected ICESat-2 RMS deviations in Figure 7 have been normalized by the radar signal wavelength to facilitate the joint analysis of the Ku- and Ka-band results. The solid line in Figure 7 represents the analytical

solution for relating the $P_c/P_n$ ratio to surface roughness via the SPM (i.e., Equation 3) (Grima et al., 2014a). It is immediately





clear that the analytical solution does not fit the observed relationship and leads to the surface roughness being almost universally overestimated. However, Figure 7 does suggest a general association between the RSR $P_c/P_n$ ratio and the wavelength-baseline projected ICESat-2 RMS deviations. Because this association linear (when plotted in log-log space), we use it to define the following empirical mapping;

$$v(\lambda)_{ICESat-2} = \lambda * \left(\frac{P_c}{P_n}\right)^{-0.892} * 10^{-1.706} \ . \tag{4}$$

The overestimation of surface roughness based on the RSR $P_c/P_n$ ratio when using the SPM may seem somewhat disconcerting; knowing its range of validity covers the smallest RMS heights ($k\sigma_h < 0.3$). However, the SPM assumes scale-independent surface roughness while, outside of very local areas (e.g., Figure 3), surface roughness across the GrIS appears strongly scale-dependent (Figures 1 and 4). As such, the SPM was likely ill-suited to the task of deriving surface roughness
from the RSR results from the beginning.

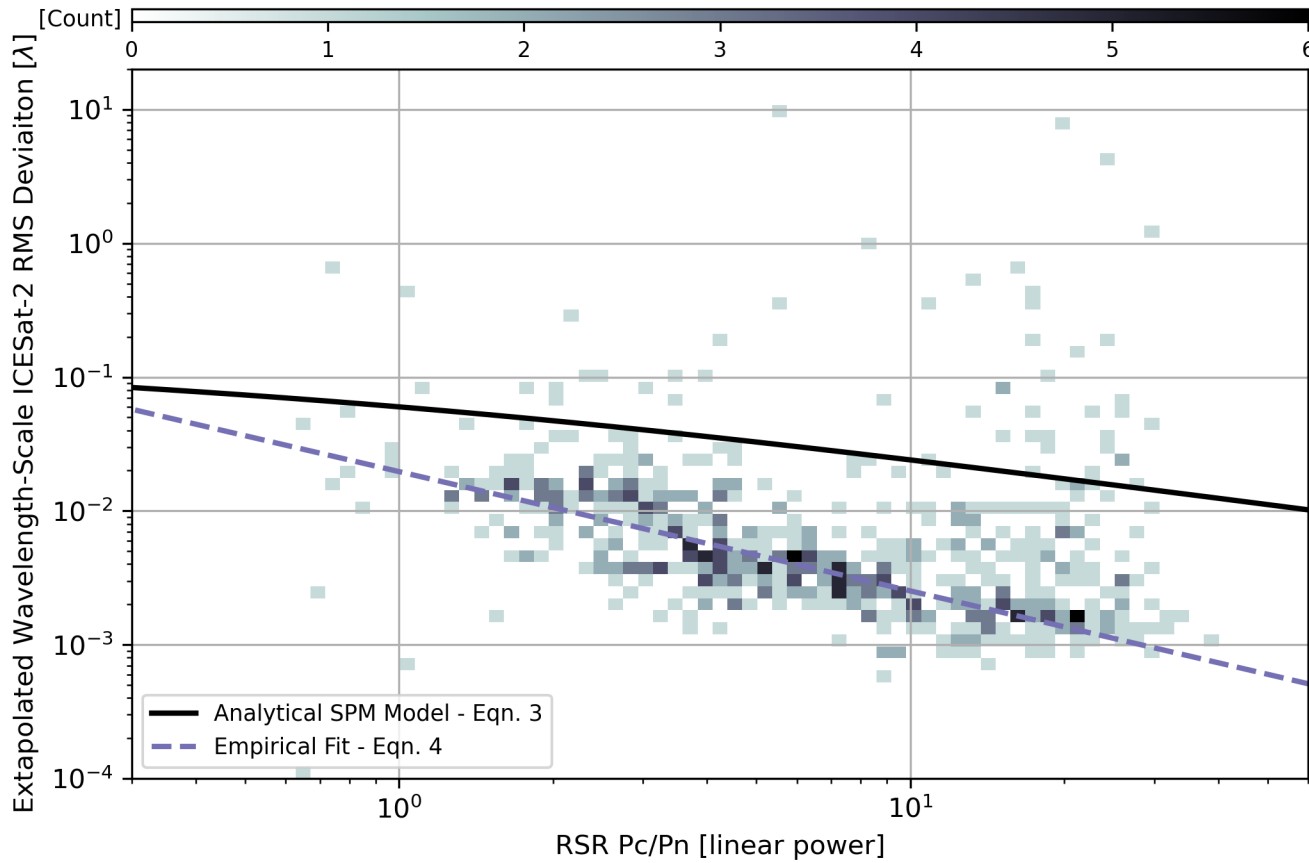

**Figure 7:** Direct comparison of the combined CryoSat-2 and SARAL coherent/incoherent ($P_c/P_n$) power ratios on which RSR estimates of surface roughness are based and the projection of the ICESat-2 RMS deviation profile between 200 and 700 m baselines to the wavelength scale. As the conventional analytical model (solid line, Equation 3) leads to overestimating surface
roughness, a new linear empirical mapping (dashed line, Equation 4) is suggested as more appropriate.



Applying this empirical relationship to deriving surface roughness estimates from the RSR outputs yields the comparison against the projected ICESat-2 RMS deviations presented in Figure 8. Comparing Figures 6b and Figure 8, surface roughness produced using the empirical mapping relation (Equation 4) clearly produces a better match than the analytical model (Equation

3). Quantitatively, the mean absolute errors between the ICESat-2 and RSR-based wavelength-normalized roughness estimates are reduced from $0.0308\,\lambda$ (CryoSat-2) and $0.0346\,\lambda$ (SARAL) for the analytical model to $0.0119\,\lambda$ and $0.0174\,\lambda$ using the empirical model; the substantial reduction indicating a much better agreement between the radar and laser roughness estimates. Furthermore, the difference between ICESat-2 and RSR-based empirical surface roughness clusters around zero for both CryoSat-2 (Figure 8b) and SARAL (Figure 8c); whereas the analytical approach led to consistently greater RSR surface

roughness. It should be noted that a similar study using a smaller number of locations in December 2018, also observed an improvement in mean absolute error using the revised empirical RSR-roughness model (Equation 4).

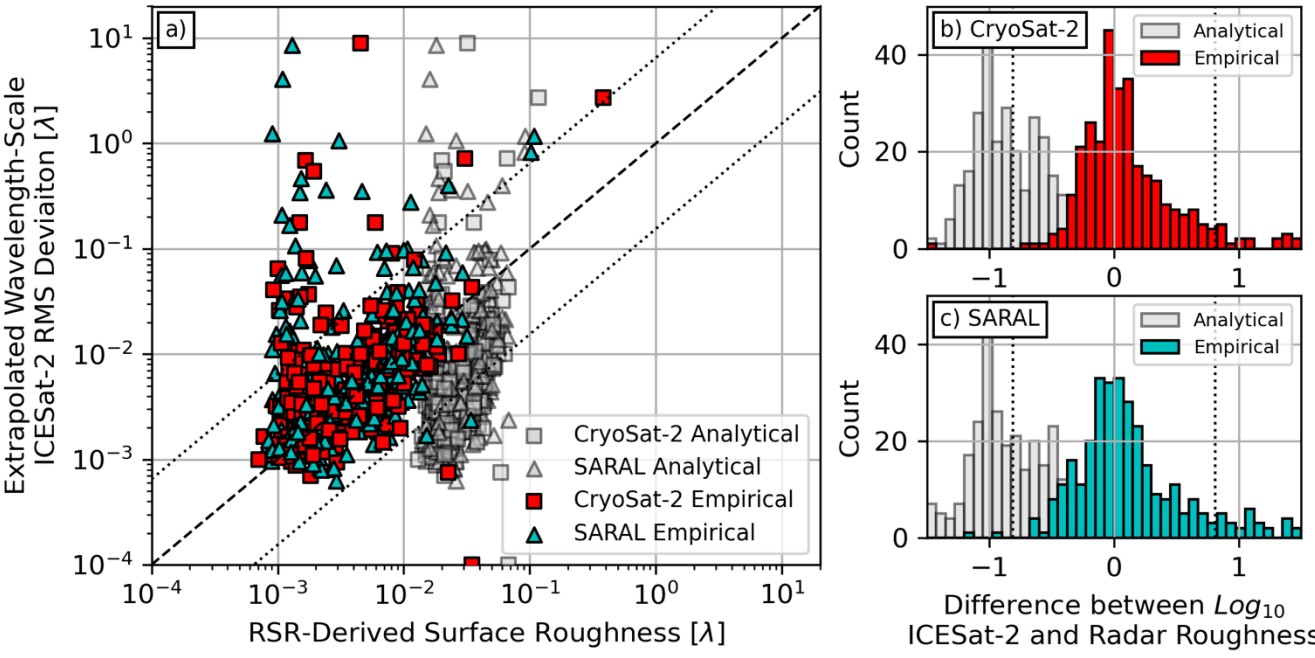

**Figure 8:** Direct comparison of the empirical and analytical RSR and ICESat-2 surface roughness results [a)] and histograms

of the differences in the logarithms of ICESat-2 and RSR surface roughness values for b) CryoSat-2 and c) SARAL. Using the empirical mapping between the RSR outputs and surface roughness, the mean absolute error is reduced to $0.0119\,\lambda$ and $0.0174$ $\lambda$ for CryoSat-2 and SARAL respectively. The analytical results are the same as those presented in Figure 6b. The dashed line in a) represents a 1:1 agreement between the radar and laser surface roughness results while the dotted lines are used to identify the $>2\sigma$ outliers in the radar and laser surface roughness estimates.




An interesting feature present in the radar/laser surface roughness comparison of Figure 8 is that substantial disagreements (i.e., $>2\sigma$ outliers) between the radar and laser altimetry surface roughness estimates are not symmetric and mainly occur above the dashed 1:1 line (i.e., an ICESat-2 surface roughness greater than that of either CryoSat-2 or SARAL). When looking at where these outliers occur spatially across the GrIS (Figures 9a and 9b), there is a clear clustering of locations in SE Greenland.

That some outlying surface roughness results can be found around the GrIS periphery or at the boundary of the different CryoSat-2 acquisition modes is not unexpected, as this is where the RSR technique is known to struggle with more spatially heterogeneous surfaces and where there are fewer data enveloping a specific location (Scanlan et al., 2023). That being said, the cluster in SE Greenland is surprising as it occurs across a high elevation and inland portion of the ice sheet. Interestingly, the SE Greenland cluster of roughness mismatches corresponds to a location where valid (i.e., quality control passing) monthly

(2013-2018) RSR results seem to be the rarest. This suggests that the GrIS surface in this area may be unique in some way that continuously affects the RSR results. Based on the ICESat-2 results from Figure 8, one possible explanation could be that this area is substantially rougher than the inland GrIS as a whole and yields distributions of radar altimetry surface echo powers that cannot be fit by a single homodyned K-distribution probability density function; thereby causing the RSR technique to fail. However, as this study focuses on understanding the RSR roughness results, a deeper assessment of the root cause for

why the RSR technique seems to experience issues in this area is left to future work.





**Figure 9:** Locations of >2σ outliers between the wavelength-scale RMS deviations projected from ICESat-2 and a) CryoSat-2 and b) SARAL surface roughness estimates. The locations are plotted on top of maps showing the number of months with valid (i.e., quality-controlled) RSR observations for the period 2013-2018 (72 months). While some outlying roughness mismatches occur closer to the boundaries of the various datasets, there is a cluster in SE Greenland that corresponds with a zone of anomalously low quality RSR results. The impact of CryoSat-2 and SARAL orbital designs can be seen in the spatial patterns (CryoSat-2 SARIn latitudinal stripping and SARAL hatching) in the southern portions of the ice sheet.

### 5.2 Spatiotemporal Patterns in Surface Roughness

Armed with an improved understanding of surface roughness derived from analysing Ku- and Ka-band satellite radar altimetry surface echoes, we can now take a closer look at the resulting spatiotemporal patterns. To this end, Figure 10 presents the 2013-2018 SARAL mean surface roughness (Figure 10a) as well as timeseries along two GrIS-bisecting transects: an east-west transect in Figure 10b and a north-south transect in Figure 10c. Blank points in the timeseries (Figures 10b and 10c)



correspond to instances where the corresponding RSR results have been removed during the quality control step (Section 3.2) and these missing data have been neglected in the calculation of the 2013-2018 mean. Note that the south-eastern portion of the GrIS, where the RSR method seems to struggle and yields results that do not meet the quality control requirements (Figure 9) can also be observed along the east-west transect as a horizontal line of missing RSR results at roughly 38º west longitude.

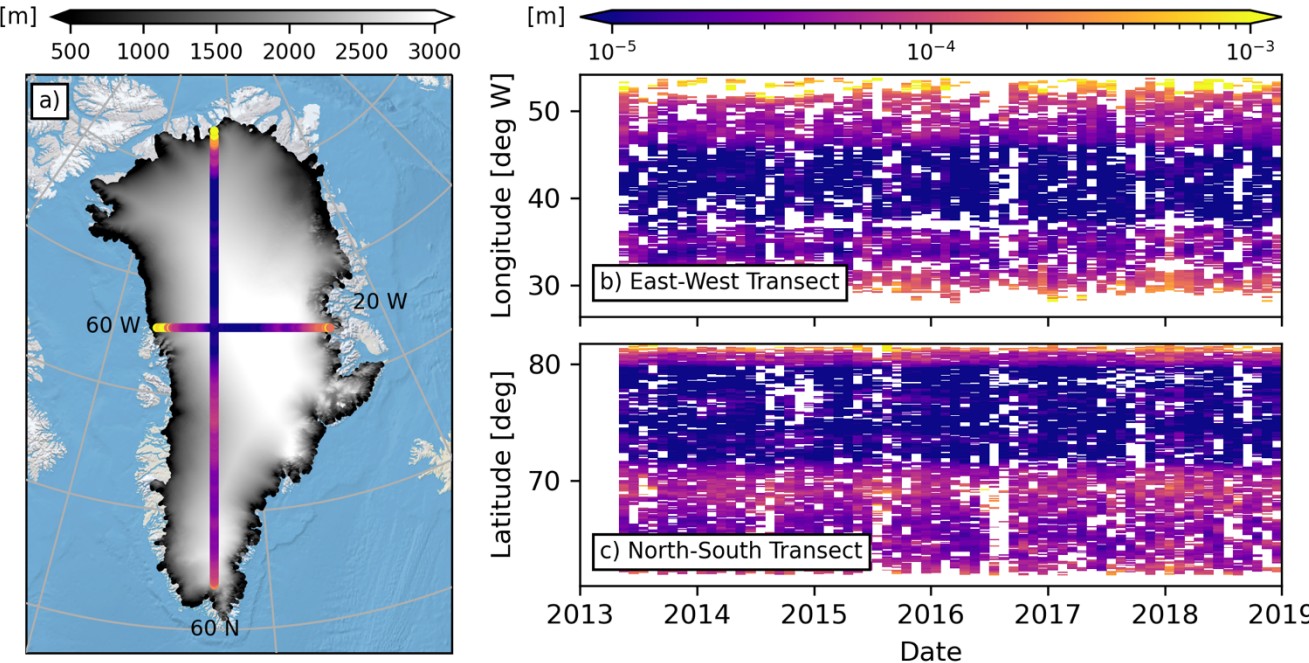

**Figure 10:** 2013-2018 SARAL surface roughness mean [a)] and timeseries [b) and c)] along east-west and north-south transects cross-cutting the GrIS. While there is strong spatial variability in RSR-derived surface roughness across the six-year period (i.e., margins are rougher than the interior), the temporal variability in surface roughness is minor. The basemap in a) is the ArcticDEM mosaic (Porter et al., 2018).

The spatial patterns in surface roughness are similar to those presented in Scanlan et al. (2023) and highlight the expected pattern of a smooth ice sheet interior that becomes progressively rougher towards the margin and in the South. The region of elevated surface roughness further inland from the margin at latitudes slightly less than 70º overlaps with the catchment of Sermeq Kujalleq (Jakobshavn Isbræ). What has changed in these new surface roughness results from those previously published is the more reliable recovery of smaller roughness values that were previously not being captured (Figure 8). Temporally, surface roughness along these transects exhibits no strong seasonal signal. There are some isolated, small variations in surface roughness (e.g., a minor increase in roughness near 69º north in mid-2015), but overall, surface roughness is strongly consistent through time. This is not surprising when considering the context for interpreting the RSR-derived surface roughness results that was established based on the CryoVEx analysis in Section 4.1. It is difficult to envision inducing rapidly repeating (e.g., annual) changes in the roughness of the GrIS surface over horizontal baselines that are hundreds of meters long



(i.e., those the RSR results seem to be projections of). Changes in roughness over these horizontal scales likely take place over longer timescales.

## 6. Relevance and Implications of RSR-Derived Surface Roughness

### 6.1 Surface Roughness by Itself

As introduced previously, surface roughness is considered an important component in SMB and heat flux modelling studies. It is typically used as an input in the calculation of either the aerodynamic roughness length or the similar drag coefficient metric. Even though it is termed a "roughness", the aerodynamic roughness length is conceptually different from how roughness is considered in the context of this study (i.e., a statistical description of the undulations in surface heights) as it quantifies the height above the ground surface at which the horizontal wind-speed profile is zero. The use of the aerodynamic roughness length and drag coefficient metric varies in practice: Jakobs et al. (2019) employs the aerodynamic roughness length as a free model tuning parameter. Amory et al. (2016) provide no direct quantifiable link between the drag coefficient and a description of surface roughness, only implying that the drag coefficient is impacted by the small-scale distribution of sastrugi. In contrast, Smeets and Van Den Broeke (2008) and van Tiggelen et al. (2023) incorporate the assumed average hummock height in their derivation of the aerodynamic roughness length across the GrIS ablation zone. In a more quantitative study based on airborne photogrammetry and spaceborne ICESat-2 laser altimetry, van Tiggelen et al. (2021) uses the standard deviation of a low-pass (<35 m wavelength) filtered, high resolution (1 m horizontal sampling) elevation profile to derive the aerodynamic roughness lengths over the K-transect. The overarching implication from all these studies is that it is the highly localized, individual surface roughness features that exert the dominating influence on the energy flux at the ice sheet surface. It is therefore the statistical descriptions of the undulations of these individual features (i.e., average height, standard deviation of heights) that then feed into GrIS SMB models.

Our validation of RSR-derived SARAL and CryoSat-2 surface roughness against CryoVEx and ICESat-2 laser altimetry shows that the assumption of scale invariant surface roughness from Scanlan et al. (2023) is ill-suited to broad regions of the GrIS. Furthermore, the RSR-derived surface roughness appears to lie below the continued projection of the RMS deviation profiles to the SARAL and CryoSat-2 radar wavelength scale (Figure 2). A fact that would not have been recognized had the CryoVEx ALS data not been used to recover RMS deviations at baselines shorter (e.g., <40 m) then are recoverable from ICESat-2 ATL06 surface heights (Figure 4). Had this analysis relied solely on ICESat-2 heights, the RSR surface roughness results would have been mistakenly interpreted as the true wavelength-scale RMS deviations (Figure 8). Instead, for the Ku- and Ka-band airborne and satellite radar altimetry data, RSR surface roughness is best interpreted as the wavelength-scale projection of the true RMS deviation profile behaviour observed between 200 and 700 m baselines. The implication is then that the SARAL and CryoSat-2 RSR surface roughness results only have physical meaning far beyond the individual roughness feature scales currently considered critical in heat flux and SMB modelling. As such, they have no direct role to play in the improvement of current GrIS SMB modelling. The relevance only at long baselines is likely also the reason why the surface



roughness timeseries presented in Figure 10 do not exhibit the strong seasonal variability that has been reported in derivations of aerodynamic roughness lengths (Smeets and Van Den Broeke, 2008; van Tiggelen et al., 2021; Van Tiggelen et al., 2023). If at some point, the scale-dependence of surface roughness that is demonstrated in this study does become more of a consideration in GrIS SMB and surface heat flux modelling, the RSR results will likely have more direct applicability.

Even though the RSR surface roughness results do not appear to be relevant as direct inputs for SMB modelling, that does not
mean they are without future intrinsic value by themselves. For example, there may be a role for directly using the radar-derived surface roughness estimates to refine the retracking of the radar waveforms and improving surface height determinations. Furthermore, there are clear spatial heterogeneities in the RSR results (Figures 8 and 9) that warrant further investigation and may shed light on the nature of GrIS surface conditions. Lastly, the ever-increasing confidence in our ability to reliably observe GrIS surface properties from CryoSat-2 and SARAL surface echo powers provides a foundation to continue
applying and adapting these techniques to earlier satellite remote sensing datasets (e.g., ERS-1, ERS-2 ENVISAT); thereby extending our observational timeseries.

## 6.2 RSR-Derived Dielectric Permittivity

Finally, and just as important as the surface roughness results themselves, revising the approach for calculating surface roughness from the coherent-incoherent power ratios will likely have on a knock-on effect on the dielectric permittivities (and
surface densities) also derived from the RSR results. This is because the impact of surface roughness is included in an adjustment term $[e^{-(2k\sigma_h^2)}]$ applied to the specular scattering equation that relates coherent power to the Fresnel reflection coefficient of the surface (Grima et al., 2012, 2014a; Scanlan et al., 2023). To that end, Figure 11 presents two-dimensional histograms comparing November 2018 analytical and empirical relative dielectric permittivity estimates from across the GrIS for each radar altimetry dataset (i.e., CryoSat-2 LRM, CryoSat-2 SARIn and SARAL). As each analytical-empirical
comparison result falls just below the dotted 1:1 line, Figure 11 highlights a slight decrease in dielectric permittivity due to the revised empirical derivation of surface roughness for each radar dataset. However, there is still a very consistent overall agreement between the two sets of permittivity estimates indicative more of a systematic adjustment in the results as opposed to a broader re-organization of spatial patterns. Individually, the variability in the analytical-empirical comparisons in Figure 11 increases from CryoSat-2 LRM (Figure 11a) to SARAL (Figure 11c) to CryoSat-2 SARIn (Figure 11b); following to the
degree to which the underlying datasets cover the rougher GrIS margin.



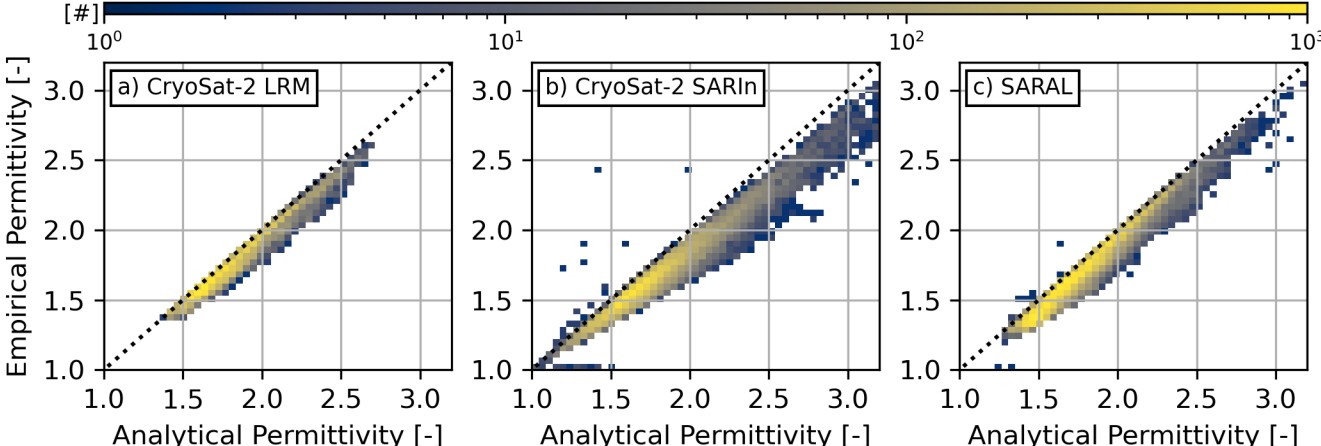

**Figure 11:** Two dimensional histograms demonstrating the impact of using the revised empirical model for calculating surface roughness from the RSR results on a) CryoSat-2 LRM, b) CryoSat-2 SARIn, and c) SARAL permittivities. Permittivities are slightly reduced in the empirical results due to the improved recovery of smaller surface roughness values.


Based on Figures 7 and 8, that the analytical permittivities are larger the empirical results is not unsurprising. For the same coherent power ($P_c$) output from the RSR analysis, a stronger Fresnel reflection coefficient (i.e., greater permittivity contrast) is required to overcome the quantitatively larger reduction to coherent power associated with overestimated analytical roughness values (Figure 8). As the empirical mapping (Equation 4) more reliably recovers smaller roughness values, the roughness adjustment applied to the RSR $P_c$ output is smaller, the corresponding permittivity contrast is reduced. That the shift in permittivity between the analytical and empirical results is small, speaks directly to the dominantly specular nature of the GrIS in terms of backscattering normal incidence Ku- and Ka-band radar signals. Now that the surface roughness results are more reliably being derived, a deeper investigation into the revised dielectric permittivities and their relevance to improving our understanding of GrIS surface density evolution will be the central focus of a follow-on study.

**7. Conclusions**

Surface roughness is an important parameter to quantify when evaluating how the Greenland Ice Sheet responds to a changing climate as it affects the efficiency of heat transfer from the atmosphere, steers meltwater, and impacts conventional measurements of ice sheet volume change. In this study, we perform a detailed investigation into a new type of surface roughness estimate derived from the Radar Statistical Reconnaissance (RSR) analysis of Ku- and Ka-band airborne (2017 and 2019 CryoVEx campaigns) and satellite (November 2018 CryoSat-2 and SARAL measurements) surface echo powers by comparing them to contemporaneous LiDAR (airborne) and ICESat-2 (spaceborne) laser surface elevations. Our results demonstrate that, outside of some specific local areas, Greenland Ice Sheet surface roughness is scale-dependent, with surface roughness increasing when quantified over larger baselines (i.e., horizontal distances) in a piece-wise pattern. It is therefore



important to consider surface roughness quantified over multiple scales. Against this backdrop, surface roughness derived
from CryoVEx radar altimetry surface echo powers do not align with the extrapolation of the LiDAR RMS deviation profiles
to the wavelength scale. In fact, they appear to align much better with the extrapolation of the piecewise linear portion of the
RMS deviation profiles for baselines between 200 and 700 meters. Building on the CryoVEx results, the direct comparison
between extrapolated ICESat-2 surface roughness RMS deviations (from the piece-wise linear portion between 200 and 700
meters) and previously published CryoSat-2 and SARAL radar surface echo powers derived using an analytical backscattering
model reveals that the radar-based results tend to overestimate surface roughness. In response, a new empirical approach is
defined to map the RSR analysis outputs (the coherent-to-incoherent power ratio) to surface roughness. The result is both and
marked improvement in the agreement between radar and laser roughness values as well as a greater dynamic range in surface
roughness across the Greenland Ice Sheet; further emphasizing the transition from a smooth ice sheet interior to a rougher
margin.

The observed sensitivity of the spaceborne RSR results to surface roughness at hundreds of meters baselines suggests they are
not well-suited to being incorporated in current surface mass balance (SMB) modelling as these models rely on the roughness
of individual meter-scale features such as hummocks or sastrugi. However, the glaciological relevance of the RSR-derived
surface roughness results is only just beginning to be understood. Future work will focus on investigating if these results can
be integrated into waveform retracking and ice sheet height estimations, detecting and mapping regional changes in ice sheet
surface behaviour, and applications to earlier remote sensing datasets to expand the current timeseries. Just as important, the
revised empirical approach for estimating surface roughness from radar altimetry surface echo powers yields a decrease in the
simultaneously derived surface permittivities; a critical piece of information for follow-on studies targeted at understanding
the observational trends in Greenland Ice Sheet surface density. Altogether, this study provides key, fundamental insight into
the derivation of Greenland Ice Sheet surface properties from radar altimetry surface echoes as well as the specific context for
how the roughness values should be interpreted.

**Acknowledgements**

KMS acknowledges that this work is carried out under the ESA Living Planet Fellowship, a programme of and funded by the
European Space Agency. KMS would also acknowledge that this work has also been supported by the ESA Network of
Resources Initiative. The procedure for recovering representative ASIRAS waveform amplitudes from the normalized
amplitudes reported in the CryoVEx data products was provided by Veit Helm.

**Code/Data Availability**

The RSR code has been retrieved from https://github.com/cgrima/rsr (accessed 7 October 2021). SARAL data products are
available through the CNES AVISO FTP (ftp-access.aviso.altimetry.fr), while CryoSat-2 data have been downloaded through



the ESA CryoSat-2 FTP (science-pds.cryosat.esa.int). The CryoVEx data used in this study are publicly available through the
CryoSat-2 cs2eo website (www.cs2eo.org). Previously published RSR results derived from CryoSat-2 and SARAL surface
echo powers (i.e., those associated with Scanlan et al., 2023) are available through http://data.dtu.dk/ with
https://doi.org/10.11583/DTU.21333291.v1.

**Author Contribution**

KMS conceptualized the study, performed the analysis as well as drafted the original manuscript and visualisations. AR and
SBS provided critical reviews and input during the revision process.

**Competing Interests**

The authors have no competing interests to declare.

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
