# Peer review of "Insights into Greenland Ice Sheet Surface Roughness from Ku-/Kaband Radar Altimetry Surface Echo Strengths"

_EGUsphere, 2024_

## Author Comment (AC1)

This manuscript is a methodical work that builds upon Scanlan et al. [2023] to pursue the layout of the foundations for using RSR-derived surface parameters at the GrIS. Eventually, it will become an entirely new type of measurement to assess the surface properties of the GrIS and the AIS. It is also the first study (all planetary bodies combined) that carefully confront RSR to altimetry roughness to better understand the baseline at which RSR roughness is derived. The manuscript is well written with a clear language and layout that both help its understanding by non-specialists in this intricated topic. The manuscript should definitely be considered for publication after the points below are addressed. The main improvements could come from extended discussions to better understand the possible causes and implications of some of the observations made. Of importance, roughness derived from laser altimetry measurements should be presented from a more critical standpoint to better understand the comparison with the RSR measurements.

**Author Response:** We'd like to begin by sincerely thanking Cyril for volunteering his time to review and comment on our manuscript. We appreciate that you find it within the scope of publication in TC and are deeply grateful for your efforts. In the following, we have gone through and responded to each of the comments raised and outlined how we have revised our manuscript in response. We believe that going through and addressing each of the comments has helped us immensely in improving the manuscript.

Our sincerest thanks once again to Cyril for taking the time to review our work and help us improve on it.

**Critic of Laser Altimetry Measurements**

The authors adopt a critical stance toward the RSR-derived roughness, which is appropriate given the relative novelty of this type of measurement, and it is in line with the goal of the manuscript to strengthen the understanding of the RSR-derived parameters for a confident use in polar science. To that aim, airborne and space-born laser altimetry are presented as reference measurements that have to be matched by the RSR for it to be validated. Laser altimetry is not necessarily deprived of possible bias, however, and this should be further discussed.

Laser- and RSR-derived roughness proceed from measurements implying different physical interactions with the surface. Laser altimetry is a discrete measurement of evenly-spaced signal delay translated into surface heights; while RSR-derived roughness fundamentally proceeds from the coherent summation of all the electric fields scattered back be each finite elements forming the surface within the radar footprint. Here are some thoughts that the authors could partly use if desired for further discussion in comparing the two measurements and explain their possible mismatch:

- What is the footprint size of the laser altimeters compared to the baseline measured? If the laser footprint is greater than the baseline, would that create a bias in the roughness derivation such as the observed on Fig.2?

- I suppose the laser altimetry heights are arranged along-track (i.e., a 1-D spatial direction) making any derived roughness highly sensitive to surface anisotropy. Conversely, I expect the RSR to be less sensitive to anisotropy as it "senses" the surface over a 2-D spatial footprint.

**Author Response:** Thank you for the comment. Cyril is correct in that we have taken the perspective of the laser altimetry results being the "truth" against which the RSR results are judged. And we agree that we have not been as critical of that assumption as is perhaps warranted. We will also raise the point that the comparison between the RSR results and the ALS versus ICESat-2 datasets are also somewhat different in that ALS is a scanning LiDAR versus the dominantly nadir pointed nature of ICESat-2. We will endeavour to address the reviewer concerns in a new sub-section to be added in Section 6.

**Actions Taken:** Please see the revisions made in Section 6. A new Section 6.3 *Revisiting Laser Altimetry as an Objective Dataset* has been added to the manuscript to address Dr. Grima's comments.

**6.3. Revisiting Laser Altimetry as an Objective Dataset**

*Throughout this study, the ALS and ICESat-2 laser altimetry data and derived surface roughness results are considered the standard against which the radar altimetry RSR-based results are assessed. However, it is also worthwhile to revisit this assumption, as the nuances in the underlying datasets may affect their relative sensitivity to surface roughness conditions.*

*First consider the scale of the footprints relative to the RMS deviation baselines. For the airborne CryoVEx data, the footprint of individual altimetry measurements are on the order of 0.7 m in diameter (ALS), 3 m along-track by 10 m across-track (ASIRAS), and 5 m along-track by 12 m across-track (KAREN) (Skourup et al., 2019, 2021). The satellite data on the other hand, as expected have larger footprint diameters ranging between 17.5 m for ICESat-2, 1.65 km for CryoSat-2 (pulse-limited footprint; 720 km altitude, 320 MHz bandwidth), and 1.4 km for SARAL (pulse-limited footprint; 800 km altitude, 500 MHz bandwidth) (Markus et al., 2017; Steunou et al., 2015; Wingham et al., 2006). As the ALS and ICESat-2 footprints are generally smaller than posting interval for their individual datasets (i.e., 1 m by 1 m for ALS, 20 m along-track for ICESat-2 ATL06), their RMS deviation profiles are assumed to be unbiased over the baselines considered (i.e., smallest roughness baseline is greater than an individual footprint). The sole exception to this though is at crossovers where the laser (ALS or ICESat-2) has sampled the same location multiple times. Examples for the ALS are T12 and T30 in 2017 and T21 in 2019. Laser data are analysed spatially as opposed to by individual CryoVEx segment or ICESat-2 orbit number, so overlaying multiple measurements on top of one another can lead to surface elevation measurements spaced less than one footprint apart. Note though that this will only affect the smallest RMS deviations*

*such as those for a 0.5 m baseline reported in Figure 2 and <20 m ICESat-2 baselines in Figure 4. At larger baselines, the individual laser surface heights will continue to be more than one footprint apart and the subsequent RMS deviation profiles should not be biased by any unaccounted for large spatial sensitivities.*

*Expanding beyond individual footprints to consider coverage, only for the CryoVEx case do the 100 m-wide ALS swaths cover the ASIRAS and KAREN radar footprints completely. In this case, it is then possible to be relatively certain that both datasets are responding to the same surface conditions. The same cannot be said for the satellite datasets. In general, the ICESat-2 surface elevations are predominantly sensitive to along-track conditions. While there are ideally surface elevations from all six of the across-track beams (subject to ICESat-2 data quality control, see Section 2.2), the spatial sampling of surface roughness will always be denser in the along-track direction. The RSR results on the other hand, represent the collected response of all scatterers within the broader illuminated radar footprint (Grima et al., 2012, 2014a, 2022). Even though attempts are made to ensure that the satellite laser and radar data being compared come from the same region (e.g., Figure 5), the sampling of the surface within those regions is not necessarily equivalent. ICESat-2 RMS deviations will be more strongly affected by any anisotropic surface conditions, whereas the CryoSat-2 and SARAL RSR results are based on a more complete two-dimensional view of the surface. Considering data on a monthly time interval further negates possible impacts of the different orbital designs and repeat cycles (91 days, 369 days, and 35 days for ICESat-2, CryoSat-2 and SARAL respectively).*

*In summary, while biases stemming from non-negligible laser footprints are considered minimal and care is taken to ensure overlapping measurements for each comparison, the different spatial footprints and sensitivities to anisotropy may still influence surface roughness derived from laser datasets and its comparison to the radar results.*

**Other Comments**

**l.129.** I understand the pulses are sumed along-track. Could you comment on how this could affect the Pc/Pn ratio and the related roughness derivation? Could that affect the mismatch between the empirical and analytical RSR roughness?

**Author Response:** Thank you for the comment. Cyril is correct in that the CryoSat-2 LRM and SARAL data have undergone along-track summing. Unfortunately, minimally processed, un-stacked waveforms are not available for these low-rate mode data products (but are for CryoSat-2 when it is operating in its SARIn mode). We agree that the implications of this on the sensitivity to surface roughness and especially the correlation length were not expressed in the initial submission. We will revise the manuscript to address this point specifically.

**Actions Taken:** Please see the revisions have been made in Section 3.2.

*Once coherent (Pc) and incoherent (Pn) are determined, deriving surface roughness requires adopting a representative backscattering model. The simplest implementation of the RSR technique (Grima et al., 2012, 2014b, a; Scanlan et al., 2023) assumes incoherent*

*backscattering from the surface follows the Small Perturbation Model (SPM) (Ulaby et al., 1982). In this case, surface RMS height can be derived following*

$$\sigma_h = \frac{\lambda e^{P_n/2P_c}}{4\pi\sqrt{P_c/P_n}},$$

*where $\lambda$ is the signal wavelength [m]. The validity bounds of the SPM are $k\sigma\_h<0.3$ and $kl<3$ where k is the radar wavenumber [m-1] and l is the surface roughness correlation length [m]. The analytical relationship for deriving RMS height from the RSR results in Equation 3 is based on the assumption that the surface roughness correlation length (i.e., the length scale over which the roughness occurs) is large relative to the radar footprint and can be neglected (Grima et al., 2012, 2014a).* **A subtle feature of this approach for both the CryoSat-2 LRM and SARAL results is that the along-track stacking inherent in the L1B and SGDR datasets (see Section 2.2) makes the RSR results more sensitive to the surface roughness correlation length in the along-track direction (Grima et al., 2014a). Stacking preferentially enhances reflections from tilted roughness elements fore and aft of the stacking midpoint, increasing their contribution to the total received power and accentuating the along-track correlation length. However, any impact due to CrySat-2 LRM and SARAL stacking is assumed to be minimal.**

*In contrast to when deriving surface dielectric permittivities from the RSR results, the absolute calibration of the RSR coherent powers is not directly required to produce a roughness estimate (Grima et al., 2012, 2014a, b; Scanlan et al., 2023). However, as an additional check on the overall applicability of the RSR approach, CryoVEx RSR results have been calibrated using the contemporaneous in-situ measurements.*

**l.185.** Could you recall to the reader why the echo is corrected from the nadir surface slope? For planetary radar sounders, this is usualy not done because the radiation pattern is not directive wrt. to the surface slope. If the transmitted signal is directive, the slope correction might work up to a certain angle depending of the beam shape (unless you also correct for the beam gain).

**Author Response:** Thank you for the comment. The large-scale slope correction comes from a preceding study (Scanlan et al., 2023) because in the very initial implementation, the RSR results were heavily influenced by the surface slope. A strong association was observed between the echo powers extracted from the satellite altimetry waveforms and the large-scale tilt of the GrIS surface. For example, a ~15 dB drop in SARAL echo power was observed for surface slopes between 0.1° and 1° (Supplementary Figure S1 in Scanlan et al., 2023). This led to a clear pattern in coherent powers which inversely matched the slope conditions (higher Pc in the interior and lower Pc around the margins). And once calibrated, the RSR inversion results were unrealistic (e.g., dielectric permittivities markedly exceeding 3.15 in the GrIS interior). Because the initial results seemed to be so dominated by larger-scale topography, the decision was made to account for it at the echo power level as opposed to at the RSR inversion stage. That being said, the reviewer's point is well taken and revisiting this decision and how it flows through to the final results would be a worthwhile exercise.

**Actions Taken:** Because this points to future work beyond the consideration of the immediate results, we have included a pointer to this specific problem when discussing future development directions at the end of Section 6.1.

*Even though the RSR surface roughness results do not appear to be relevant as direct inputs for current SMB modelling, that does not mean they are without future intrinsic value by themselves.* **First,**  *there may be a role for directly using the radar-derived surface roughness estimates to refine the retracking of the radar waveforms and improving surface height determinations.* **Second** *, there are clear spatial heterogeneities in the RSR results (Figures 8 and 9) that warrant further investigation and may shed light on the nature of GrIS surface conditions.* **Third, they represent a baseline for interpreting the RSR results in the context of other radar backscattering models (Fung and Chen, 2004; Ulaby et al., 1982) and revisiting some of the underlying decisions (e.g., correcting echo powers for nadir slopes prior to RSR processing).** *Lastly, the ever-increasing confidence in our ability to reliably observe GrIS surface properties from CryoSat-2 and SARAL surface echo powers provides a foundation to continue applying and adapting these techniques to earlier satellite remote sensing datasets (e.g., ERS-1, ERS-2, ENVISAT); thereby extending our observational timeseries.*

**l.207.** Maybe recall that this condition on the correlation coefficient is arbitrary, bit it enables to discard some of those terrains with more than one roughness regime in order to comply with the assumption behind the HK statistics.

**Author Response:** Thank you for the comment. We agree that the assumptions implicit in the definition of "quality" when it comes to assessing the RSR results are not being communicated as clearly as they should be. We will revise the manuscript to make this more explicit.

**Actions Taken:** Please see the additions made in Section 3.2.

*Two metrics are used to quality control the RSR results: first, the distance to the furthest surface echo power measurement considered, and second, the correlation coefficient between the observed surface echo power histogram and the statistical fit. The former is irrelevant for the CryoVEx data as we mandate using the closest 1000 data points surrounding the in-situ locations regardless of how far they may be. For the satellite results, search radii of 50 km, 25 km, and 40 km are used for the CryoSat-2 LRM, CryoSat-2 SARIn, and SARAL results respectively (Table 1). The CryoSat-2 LRM and SARAL maximum search radii are the same as those used in Scanlan et al., (2023), while the CryoSat-2 SARIn radius is extended as more datapoints are being considered. A minimum correlation coefficient of 0.96 is required for all RSR results (Grima et al., 2012, 2014b, a; Scanlan et al., 2023).* **This threshold is used to select only locations where the envelope of observed surface echo powers can be well-described by the homodyned K-distribution, which includes implicit assumptions for how radar energy is scattered from the surface (Grima et al., 2014a). As such, locations that do not meet this correlation coefficient threshold are not necessarily of poorer quality but require a different interpretation of the observed echo powers. This study though focuses solely on locations well-described by the homodyned K-distribution statistics.**

In addition to revisions made in Section 3.2, we have also modified the language in the Figure 9 caption to better reflect the nuances in the quality control assessment.

*Figure 9: Locations of >2σ outliers between the wavelength-scale RMS deviations projected from ICESat-2 and a) CryoSat-2 and b) SARAL surface roughness estimates. The locations are plotted on top of maps showing the number of months with valid (i.e., quality-controlled) RSR observations for the period 2013-2018 (72 months). While some outlying roughness mismatches occur closer to the boundaries of the various datasets, there is a cluster in SE Greenland at ~3000 m elevation in the vicinity of the ice divide that corresponds with a zone of RSR results **that do not meet the quality control criteria**. The impact of CryoSat-2 and SARAL orbital designs can be seen in the spatial patterns (CryoSat-2 SARIn latitudinal stripping and SARAL hatching) in the southern portions of the ice sheet.*

**Fig.2.c-d and the related discussion.** I'm intrigued about this scale-invariant roughness for the laser. Would it be possible for it to be a bias due to the technique? This mode happend for very low baselines (< 10m). How does this compare with the laser footprint?

**Author Response:** Thank you for the comment. This is related to one of the main points Cyril raised previously, so please see our response above. However, we do not think there is a footprint-introduced bias in the ALS data as the stated footprint is less than one meter at the nominal flying altitude of 300 m. So small baselines down to one meter should be reliably resolved by the laser scanner. The seemingly scale invariant roughness at the T30 and T41 seems to be a highly localized feature unique to these locations.

**Actions Taken:** Please see the revisions made in response to the reviewer's previous 'Critic of Laser Altimetry Measurements' comment.

**l.251-53.** Could you discuss this observation? I wonder if there is a physical sense that could explain that this 200-700m baseline is better for radar roughness downscaling. Is this baseline range related to the radar footprint somehow? This is the tricky part for the radar. Any individual scattering is usually associated to a wavelength-scale baseline (although this is a just common theoretical assumption), but the coherent wave front recorded at the antenna is the summation of all those scatteres from within a much wider footprint. In the end, the scale of the footprint migth also have a role in the derived roughness.

**Author Response:** Thank you for the comment. The radar footprint radii range from meter-scale for the airborne CryoVEx case to ~1.5 km for the satellites (i.e., CryoSat-2 and SARAL). All the while the alignment with the 200-700 m ICESat-2 roughness baseline seems to be consistent for both the airborne and satellite cases.

Why the RSR results seem to be keying off that 200-700 m interval is something we have thought long about but have yet to come to a firm conclusion. But it is important to consider that focusing too much on 200-700 m interval specifically may be a bit of a misnomer. That specific baseline range is taken because it is where we believe we have both reliable ALS and

ICESat-2 RMS deviation (Allan) profiles (e.g., Figure 4). This is because the along-track ICESat-2 ATL06 data cannot provide insight at baselines shorter than the 20 m posting interval and we only consider ALS data within 1 km of the in-situ locations (capping maximum investigable baseline).

If we take a closer look into some of the individual ICESat-2 Allan profiles (two more examples shown below), we can see that the linear behaviour observed between 200 and 700 m can extend well beyond this range, to baselines of ~3 km (at which there seems to be another breakpoint). Here the baselines are on the order of the CryoSat-2 and SARAL footprint diameters but are substantially greater than the airborne CryoVEx footprints. Note also, if we push the ICESat-2 data to as short of baselines as possible using data near orbit intersections (hence there are only a small number of comparative data points; black solid and dashed lines in the figures below), we do see that the RMS deviations start to level off a bit similar to what is observed in the ALS data.

[Figure]

All this being said, we agree that some of the nuances around the 200-700 m interval need to be addressed more explicitly in the manuscript. We have incorrectly implied it to be a 'fixed' interval, while it is better thought of as a consequence related to the nature of the datasets we are looking at. While we cannot tie the apparent sensitive roughness scale concretely to the radar footprint (the CryoVEx footprints are much smaller), we can state that based on the observations, the minimum consistent baseline the RSR results appear sensitive to is on the order of 100's of meters. We will make numerous revisions in the manuscript to try and make this clearer.

**Actions Taken:** We have revised the ICESat-2 RMS deviation profiles in Figure 4 to illustrate some of the more nuanced aspects of these data, especially when pushing the data as far as possible at small baselines.

[Figure]

The Figure 4 caption has been modified as follows.

*Figure 4: The comparison of March/April 2017 ALS (blues) and November 2018 ICESat-2 (greens) RMS deviation profiles centred on locations T30 (light) and T41 (dark) along the EGIG line. There is a substantial discrepancy between the two sets of RMS deviation profiles in the overlapping baseline range (100 m to 1 km) further confirming that the local regions surrounding T30 and T41 are markedly smoother than those further afield. The anomalous behaviour in the ICESat-2 RMS deviation profiles at **both small (i.e., <80 m) and long**  (i.e., >4 km) baselines is related to the quick drop-off in the number of comparable surface elevations.*

In-line with the revision of Figure 4, the following revisions have been made in Section 4.1.

*Amongst the ALS results, the 2017 T30 and T41 RMS deviation profiles (Figure 2c) are unique, in that they do not exhibit the increased scale-dependent roughness behaviour at longer baselines. Instead, their RMS deviation profiles are flat and monotonic (i.e., not piecewise linear). The reason for this change in surface roughness behaviour is due to the ALS data surrounding T30 and T41 preferentially covering extremely smooth local areas of the GrIS. Figure 3 presents the local surface elevations (Figures 3a and 3b) along with the height deviations (elevations minus the constant location-specific background plane; Figures 3c and 3d) surrounding the T30 and T41 ALS datasets. Elevations are taken from the 10 m ArcticDEM mosaic (Porter et al., 2018) and the background plane is defined using all 10 m ArcticDEM data within 20 km of the CryoVEx in-situ measurement location. It is clear that the topographic variability across each of these sites is very small at T30 and essentially non-existent at T41. It is then not surprising that the corresponding RMS deviation profiles (Figure 2c) do not exhibit the increase in RMS deviation at large horizontal baselines that is observed at the other CryoVEx locations. To further emphasise the smoothness of the GrIS near T30 and T41, Figure 4 compares the ALS RMS deviation profiles with those derived from all ICESat-2 surface elevations within 25 km and 35 km of T30 and T41, respectively. We must use ICESat-2 data that are further away from the T30 and T41 sites because these locations are between ICESat-*

*2 orbital ground tracks. When considering surface topography over a broader regional area (i.e., ICESat-2), the stronger scale dependency in surface roughness is once again observed. Less scale dependency exists in the ALS RMS deviation profiles between 100 m and 1 km because T30 and T41 are sited in a locally very smooth portion of the GrIS (Figure 3). *

*In addition to the return of scale-dependent roughness when considering the broader regions surrounding T30 and T41, the ICESat-2 results in Figure 4 also demonstrate two other notable points. The first is that when the ICESat-2 baseline is pushed to its shortest limit (i.e., considering closely spaced surface elevations at orbit crossovers), the RMS deviation profile appears to flatten. While there are only a small handful of surface elevation measurements at these short baselines (e.g., 10's of points separated by 10 m compared to 10,000's of points separated by 20 m), the ICESat-2 results do seem to exhibit the same less scale-dependent roughness pattern as has been observed in the ALS results (Figures 2c and 2d). And again, as with the ALS data, the transition in the RMS deviation profile occurs in the 100 m range. The second notable point is that the piecewise linear, scale-dependent behaviour in the RMS deviation observed in the ALS data between 200 and 700 m baselines (Figures 2c and 2d) appears to continue well beyond that range, to upwards of roughly 3 km in the ICESat-2 results. Recall that only ALS data within 1 km of the in-situ measurement location underlie the RMS deviation profiles in Figure 2. The 200 to 700 m interval used in the projection of the RMS deviation profiles to the radar wavelengths is used solely because it is an interval common to both the ALS and ICESat-2 results. The scale-dependent behaviour in the surface roughness appears to extend to much longer baselines. Lastly, while the ICESat-2 RMS deviation profiles to being to suffer data availability issues, there does appear to be another marked transition to less scale-dependent roughness at the longer (i.e., >4 km) baselines.*

The following revision has been made in Section 4.2.

*In lieu of presenting hundreds of individual ICESat-2 RMS deviation profiles together with the CryoSat-2 and SARAL RMS heights from Scanlan et al. (2023) (i.e., akin to Figures 2c and 2d) and following on from what has been learned from the CryoVEx results, Figure 6 presents the **comparison** of the RSR RMS heights, and the wavelength-scale RMS deviations projected from a linear fit to the RMS deviation profiles for baselines between 200 and 700 m for 328 locations across the GrIS. **As presented in relation to Figure 4, the 200-700 m interval is selected due to its general representativeness of the piecewise linear, scale-dependent roughness behaviour observed in both the ALS and ICESat-2 data. It should not be considered a uniquely fixed interval.** These 328 locations have been pseudo-randomly selected based solely on considerations for the computational load when performing the point-to-point surface deviation comparison as part of the RMS deviation profile calculation (i.e., ≤55,000 ICESat-2 surface elevations). To ease the comparison, all surface roughness estimates*

*(i.e., CryoSat-2 and SARAL RSR RMS heights or ICESat-2 RMS deviations projected to the CryoSat-2 and SARAL wavelength scale) have been normalized by the radar signal wavelength. While there may be the suggestion of possible linear relationship between the radar- and laser-derived surface roughness estimates, the is clearly no 1:1 agreement. This appears to be in part due to a floor in the RSR results as they consistently fail to recover the smallest ICESat-2 RMS deviations. The mean absolute error between the two sets of surface roughness estimates is 0.0308 λ for CryoSat-2 and 0.0346 λ for SARAL.*

The following changes have been made in Section 6.1.

*Instead, for the Ku- and Ka-band airborne and satellite radar altimetry data, RSR surface roughness is best interpreted* **not as the true wavelength-scale RMS deviation, but the projection of the scale-dependent behaviour observed at baselines between hundreds of metres and a few kilometres to the wavelength scale** *. The implication is then that the SARAL and CryoSat-2 RSR surface roughness results only have physical meaning far beyond the individual roughness feature scales currently considered critical in heat flux and SMB* **studies***. As such, they have no direct role to play in the improvement of current GrIS SMB modelling. The relevance only at long baselines is likely also the reason why the surface roughness timeseries presented in Figure 10 do not exhibit the strong seasonal variability that has been reported in derivations of aerodynamic roughness lengths (Smeets and van den Broeke, 2008; van Tiggelen et al., 2021, 2023).*

The following revisions have been made in the Section 7.

*Against this backdrop, surface roughness derived from CryoVEx radar altimetry surface echo powers do not align with* **a continuation** *of the LiDAR RMS deviation profiles to the wavelength scale. In fact, they appear to align much better with the extrapolation of* **the consistent** *piecewise linear portion of the RMS deviation profiles* **observed at** *baselines between* **hundreds of metres and a few kilometres***. Building on the CryoVEx results, the direct comparison between extrapolated ICESat-2 surface roughness RMS deviations (from the piece-wise linear portion between 200 and 700 meters) and previously published CryoSat-2 and SARAL radar surface echo powers derived using an analytical backscattering model reveals that the radar-based results tend to overestimate surface roughness.*

*The observed sensitivity of the spaceborne RSR results to surface roughness* **that is at its smallest, hundreds of meters in scale** *suggests they are not well-suited to being incorporated in current surface mass balance (SMB) modelling as these models rely on the roughness of individual meter-scale features such as hummocks or sastrugi.*

**l.310.** Throughout the paper, the RMS deviation (aka. Allan deviation) and RMS heigth appears to be used interchangeably. I understand the authors do know they are different, but it is not clearly highligthed in the text and the reader could be misled, especially when those two parameters are compared on the same 1:1 plot (Fig.6b). A

brief discussion on the difference between RMS deviation and RMS heigth should be added to understand if one should expect any bias in that figure or not.

**Author Response:** Thank you for the comment. Confusion is understandable and is a result of imprecision on our part. The RMS heights presented here are those coming directly from the previous publication (i.e., Scanlan et al., 2023) where the SPM was used in the inversion of the RSR results. We will check the manuscript to make sure the context is clear whenever specific terminologies are used. We will also include a brief description of the RMS height and its main salient difference compared to the RMS deviation (i.e., no consideration of scale).

**Actions Taken:** The following revision has been made in Section 3.2.

*In this case, surface RMS height can be derived following*

$$\sigma_h = \frac{\lambda e^{P_n/2P_c}}{4\pi\sqrt{P_c/P_n}}, \qquad (3)$$

*where $\lambda$ is the signal wavelength [m].* **In contrast to RMS deviation (Equation 1), RMS height is simply the standard deviation of the surface heights (after removing the mean) with no consideration of horizontal scale (Shepard et al., 2001).** *The validity bounds of the SPM are $k\sigma\_h<0.3$ and $kl<3$ where k is the radar wavenumber [m-1] and l is the surface roughness correlation length [m].*

The following revision has been made in Section 4.2.

*In lieu of presenting hundreds of individual ICESat-2 RMS deviation profiles together with the CryoSat-2 and SARAL RMS heights* **from Scanlan et al. (2023)** *(i.e., akin to Figures 2c and 2d) and following on from what has been learned from the CryoVEx results, Figure 6 presents the* **comparison** *of* **those initial** *RSR RMS* **height estimates from Equation 3** *and the wavelength-scale RMS deviations projected from a linear fit to the RMS deviation profiles for baselines between 200 and 700 m for 328 locations across the GrIS.*

We have revised the horizontal axis label of Figure 6 to more clearly express the provenance of these data.

[Figure]

The following revision has been made to the Figure 6 caption as well.

*Figure 6: Panel a) presents the locations of 328 pseudo-randomly chosen locations across the GrIS where in Panel b)* **Scanlan et al. (2023)** *RSR surface roughness results (CryoSat-2 as squares and SARAL as triangles) are compared to wavelength-baseline projected RMS deviations from ICESat-2. While there is a general positive associated between the two sets of roughness estimates, the RSR results do not reliably recover the smallest ICESat-2 roughness levels. The mean absolute error between the ICESat-2 and the CryoSat-2 and SARAL RSR-based wavelength-normalized roughness estimates are 0.0308 λ and 0.0346 λ respectively.*

**Eq.4.** Although the methodology to obtain such relationship could be used elsewhere, its coefficients migth not. It is then necessary to discuss the perceived validity domain of this equation to avoid the community to use it with any radar system and in any environment.

**Author Response:** Thank you for the comment. We agree that, this point should be made explicit in the manuscript. We will revise the manuscript to include it.

**Actions Taken:** The following revisions have been made in Section 5.1.

*Applying this empirical relationship to deriving surface roughness estimates from the RSR outputs yields the comparison against the projected ICESat-2 RMS deviations presented in Figure 8. Comparing Figures 6b and Figure 8, surface roughness produced using the empirical mapping relation (Equation 5) clearly produces a better match than the analytical model (Equation 3). Quantitatively, the mean absolute errors between the ICESat-2 and RSR-based wavelength-normalized roughness estimates are reduced from 0.0308 λ (CryoSat-2) and 0.0346 λ (SARAL) for the analytical model to 0.0119 λ and 0.0174 λ using the empirical model; the substantial reduction indicating a much better agreement between the radar and laser roughness estimates. Furthermore, the difference between ICESat-2 and RSR-based empirical*

*surface roughness clusters around zero for both CryoSat-2 (Figure 8b) and SARAL (Figure 8c); whereas the analytical approach led to consistently greater RSR surface roughness.* **A** *similar study* **only with** *a smaller number of locations in December 2018, also observed an improvement in mean absolute error using the revised empirical RSR-roughness model (Equation 5).* ***Expanding more broadly, the form of Equation 5 and the procedure for developing it could be adapted to other RSR implementations on Earth as well as beyond, but care will have to be taken to ensure the coefficients are appropriate as they may vary in different contexts/applications.***

**Fig.8.** It sounds too me there is also another outlier group of points between 0.01. and 0.1m, just below the bottom dotted line. Does this group of points have a spatial cohesivness?

**Author Response:** Thank you for the comment. The locations plotted in Figure 9 are not discriminated between those above 2-sigma or below, all outliers are plotted together. We see now that should have been communicated more clearly. We expect most of the clustering in SE Greenland will be the >2-sigma outliers since there are just more of them. There does not seem to be any obvious cluster that is not on a data coverage boundary besides that in the southeast.

**Actions Taken:** The following revision has been made in Section 5.1.

*An interesting feature present in the radar/laser surface roughness comparison of Figure 8 is that substantial disagreements (i.e., >2σ outliers) between the radar and laser altimetry surface roughness estimates are not symmetric and mainly occur above the dashed 1:1 line (i.e., an ICESat-2 surface roughness greater than that of either CryoSat-2 or SARAL). When looking at where* **all of** *these outliers occur spatially across the GrIS (Figures 9a and 9b), there is a clear clustering of locations in SE Greenland. That some outlying surface roughness results can be found around the GrIS periphery or at the boundary of the different CryoSat-2 acquisition modes is not unexpected, as this is where the RSR technique is known to struggle with more spatially heterogeneous surfaces and where there are fewer data enveloping a specific location (Scanlan et al., 2023).*

We have also made the following revision to the Figure 9 caption.

*Figure 9: Locations of* **all** *>2σ outliers between the wavelength-scale RMS deviations projected from ICESat-2 and a) CryoSat-2 and b) SARAL surface roughness estimates. The locations are plotted on top of maps showing the number of months with valid (i.e., quality-controlled) RSR observations for the period 2013-2018 (72 months). While some outlying roughness mismatches occur closer to the boundaries of the various datasets, there is a cluster in SE Greenland at ~3000 m elevation in the vicinity of the ice divide that corresponds with a zone of RSR results that do not meet the quality control criteria. The impact of CryoSat-2 and SARAL orbital designs can be seen in the spatial patterns (CryoSat-2 SARIn latitudinal stripping and SARAL hatching) in the southern portions of the ice sheet.*

**l.380-84.** The authors suggestion is completly valid. I would like to propose additional explanations that could change the Pc/Pn ratio used to derive the radar roughness. The authors are free to consider them or not:

- A differential volume scattering between two terrains will also affect the Pc/Pn ratio. Volume scattering will add incoherent energy.

- If there is a strong roughness anysotropy, the along-track measurements of laser altimetry migth differ from the radar sensitive to more spatially distributed scatterers.

- Thin layering (firn crust?) could change the coherent energy.

- When the surface is too rough, the surface mean slopes will start to scatter most of the energy off-nadir. The Pn measured at the antenna is then an underestimate of the integrated incoherent energy arising from that surface

**Author Response:** Thank you for the comment. First off, re-reading these sentences and in light of some of Dr. Grima's other comments, we do think the language here needs to be cleaned up. Second, we think these are great alternative suggestions that could explain the anomalous results in this SE portion of the GrIS.

**Actions Taken:** Please see the revisions …

*That being said, the cluster in SE Greenland is surprising as it occurs across a high elevation and inland portion of the ice sheet. Interestingly, the SE Greenland cluster of roughness mismatches corresponds to a location where*  *monthly (2013-2018) RSR results* **meeting the quality control criteria (Section 3.2) are amongst**  *the rarest. This suggests that the GrIS surface in this area may be unique in some way that continuously affects the RSR results. Based on the ICESat-2 results from Figure 8, one possible explanation could be that this area is substantially rougher than the inland GrIS as a whole and yields distributions of radar altimetry surface echo powers that cannot be* **cleanly** *fit by a single homodyned K-distribution probability density function*. **Other alternative explanations that could incite changes in the distribution of surface echo powers include a local variation in volume scattering affecting the amount of diffuse scattering and firn crusts/thin layering affecting the specular component.** *However, as this study focuses on understanding the RSR roughness results, a deeper assessment of the root cause for why the RSR technique seems to experience issues in this area is left to future work.*

**l.391.** The RSR is not especially lower quality, it is indicative of some sort of additional surface properties that is not caught by the laser (they are hard to disentangle, though) as mentioned in l.457-58.

**Author Response:** Thank you for the comment. We agree that this was a poor choice of words on our part. Along with the Dr. Grima's earlier comment on what the 0.96 correlation

coefficient actually implies about the results, we will revise the manuscript to try and make this clear.

**Actions Taken:** The following revisions have been made to the Figure 9 caption.

*Figure 9: Locations of >2σ outliers between the wavelength-scale RMS deviations projected from ICESat-2 and a) CryoSat-2 and b) SARAL surface roughness estimates. The locations are plotted on top of maps showing the number of months with valid (i.e., quality-controlled) RSR observations for the period 2013-2018 (72 months). While some outlying roughness mismatches occur closer to the boundaries of the various datasets, there is a cluster in SE Greenland at ~3000 m elevation in the vicinity of the ice divide that corresponds with a zone of*  *RSR results **that do not meet the quality control criteria**. The impact of CryoSat-2 and SARAL orbital designs can be seen in the spatial patterns (CryoSat-2 SARIn latitudinal stripping and SARAL hatching) in the southern portions of the ice sheet.*

**l.481.** larger "than".

**Author Response:** Thank you for the comment. This was a typo that made it through our proof-reading and will be corrected

**Actions Taken:** Please see the revision has been made in Section 6.2.

*Based on Figures 7 and 8, that the analytical permittivities are larger **than** empirical results is not unsurprising.*

---

## Author Comment (AC2)

**Review of "Insights into Greenland Ice Sheet Surface roughness from Ku-/Ka-band Radar Altimetry Surface Echo Strengths" submitted to The Cryosphere in 2024 by Kirk M. Scanlan et al.**

This study focuses on the quantification of surface topographic roughness of the Greenland ice sheet using CryoSat-2 and SARAL satellite radar measurements. The Radar Statistical Reconnaissance (RSR) method, developed in a recent publication by the same authors, is used for this purpose. The derived roughness properties are compared against airborne radar and laser scanner data from two campaigns, and with ICESat-2 ATL06 (20m intervals) data. The main question is whether the surface roughness can be accurately estimated using spaceborne radar, the main motivation being that the surface roughness is an important yet poorly constrained parameter for SMB models. This makes this study relevant for The Cryosphere. This study is also original, as novel insights are presented regarding the estimation of surface roughness using satellite radar datasets, such as the empirical correction for the RSR method (Eq. 4) that gives a better match with ICESat-2 data, and the unique spatio-temporal analysis of surface roughness (Fig. 10), which form a basis of future work using such data. The potential significance of this work is also high, since improved estimation of ice sheet roughness properties allow not only for potential improved SMB modelling but also improved radar postprocessing. The employed datasets and methods are sound, as far as I can assess as a non-radar expert.

Overall, the paper is well written. It makes use of adequate referencing from the remote sensing and SMB modelling perspectives, although it could benefit from more referencing in the field of the snow bedform quantification. Some examples are given below. The introduction allows for the non-remote sensing expert to understand the overall topic. However, the methods and results sections are long and, in some places in the methods, technical terms are introduced which do not appear to be very relevant for the results. On the other hand, there is no mention on how the laser scanner and ICESat-2 data were processed and filtered, even though these data are used as reference. The results are very interesting, yet I have some concerns which are listed below. The discussion is a relevant and a welcome part as it presents the main limitation of the novel results for improved SMB modelling. However, some more work is required to make this radar roughness estimation useful for SMB modeling, which I recommend should be done or at least discussed more quantitively before final publication. Is there absolutely no way to use spaceborne radar for improved roughness estimation relevant for SMB models ? At present a constant aerodynamic roughness value is still chosen over the entire ice sheet in climate models.

To summarise, I would recommend publication after the main comments are clarified, and once the derived roughness can be (indirectly) used for SMB modelling.

**Author Response:** We'd like to begin by sincerely thanking the anonymous reviewer for volunteering their time to review and comment on our manuscript. We are deeply grateful and appreciative of their efforts. In the following, we have gone through and responded to each of the comments raised and outlined how we have revised our manuscript in response. We believe

that going through and addressing each of the reviewer's comments has helped us immensely in improving the manuscript.

Our sincerest thanks once again to the reviewer for taking the time to review our work and help us improve on it and its relevance to TC.

**Major comments**

This study uses an airborne laser scanner (ALS) and ICESat-2 ATL06 data at 20m resolution as independent reference data. First of all, it is not clear what the accuracy of the ALS data is, and how the 1m effective ALS resolution (L114) relates to the resolution required to detect the relevant surface features in the areas of interest (sastrugi, snow dunes, etc…). Even with ~1m resolution and ~10cm vertical accuracy, the ALS is limited in capturing all the scales relevant for SMB modelling, which is the main aim of this work (according to the introduction). Therefore, I would invite the authors to also critically assess the ALS data in relation to the physical processes of interest.

**Author Response:** Thank you for the comment. The vertical accuracy of the ALS system was mistakenly omitted from the manuscript. As the reviewer rightly points out, this accuracy is on the order of 10 cm based on cross-over differences (Skourup et al., 2019, 2021). This will be included in the revised version along with a brief summary of some of the more salient points related to ALS processing as suggested by the reviewer in one of their minor comments below. We also realize that it is not clear in the manuscript if we performed the processing of the CryoVEx data or if it was done beforehand. The data we collected from ESA (cs2eo.org) are already processed, so none of what is described in Section 2.1 are things we have implemented for this study but are features of the available datasets.

Regarding the latter comment, at it's core, what we want to get out of this study is the correct way to interpret the roughness measurements derived from the RSR analysis of CryoSat-2 and SARAL surface echo powers. Why we believe this is valuable is because 1) we'll then have a tractable long-term monthly record of surface roughness across the entire Greenland Ice Sheet and 2) based on that interpretation, our roughness results may be relevant for SMB modelling. Our approach is to rely on contemporaneous airborne and satellite laser altimetry datasets to establish the correct framing of the RSR results, not to assess the viability of every surface roughness dataset for SMB modelling. However, we agree that by the end of the manuscript, we will have analysed different datasets and are in a position to comment on their relevance to SMB modelling. We will revise Section 6.1 to also include a broader statement on the relevance of all the altimetry-based roughness results.

**Actions Taken:**

In reference to the reviewer comment on ALS accuracy and data processing, the following change has been made in Section 2.1.

*This study uses March/April 2017 (Skourup et al., 2019) and April 2019 (Skourup et al., 2021) CryoVEx data (ESA, 2022a, b) collected along the EGIG line in central Greenland. Specifically, it focuses on ASIRAS, KAREN, and ALS measurements surrounding positions of*

*the in-situ measurements performed at the T5, T9, T12, T19, T30, and T41 locations in 2017 as well as T9, T12, T21, and T35 locations in 2019.* **All CryoVEx datasets are available having already undergone thorough data processing (Skourup et al., 2019, 2021).** *Both the ASIRAS and KAREN data have been subject to post-acquisition SAR (synthetic aperture radar) processing to minimise the size of their respective footprints in the alongtrack (3 m and 5 m, respectively) and crosstrack (10 m and 12 m, respectively) directions. The ALS data have been* **calibrated, had outliers removed, and organized**  *into 200-300 m wide swaths following the aircraft ground track*. **Individual ALS** *ground height measurements* **are spaced at one-meter by one-meter intervals** *and have a vertical accuracy of* **~10cm (Skourup et al., 2019, 2021)**.

The following new paragraph has been added in Section 6.1 to comment more broadly on the relevance of the altimetry-based surface roughness results we are seeing through our analysis.

 **Taking a broader view of all the airborne and satellite altimetry (i.e., radar and laser) datasets analysed in this study, they all point to GrIS surface roughness being, at least for some baselines (e.g., 200 m to 3 km), scale dependent. While there are isolated local areas where roughness is not strongly dependent on the horizontal distance over which it is measured (e.g., immediately surrounding T30 and T41 in Figures 2-4), when considering the broader regional conditions, scale dependent roughness appears commonplace (as implied by Figures 7 and 8). This challenges current SMB models that incorporate surface roughness via a single spatially homogeneous value (e.g., the aerodynamic roughness length). An avenue for future research is then to integrate the effects of scale-dependent roughness into numerical SMB and heat flux models and assess their predictions of ice sheet evolution against reality. At this point, the roughness results derived from the RSR analysis of CryoSat-2 and SARAL surface echo powers will likely have more direct applicability. It should be noted that quantifying surface roughness at the scale of individual features (e.g., sastrugi, hummocks) will be challenging for standard satellite measurements (e.g., ICESat-2 ATL06 land heights, Figure 4) and integration with more specialized analyses or alternative datasets (e.g., LiDAR, UAV photogrammetry, etc.) are likely to be required (van Tiggelen et al., 2021).**

Similarly, at 20m resolution and >10cm vertical accuracy, the ICESat-2 ATL06 will evidently be even more  limited in quantifying roughness features such as sastrugi and all the way down to the radar wavelength scale. Hence, I am skeptical about the interpretation of the empirical correction to the RSR method in Eq. 4. I would argue that this is an empirical or "data-based" correction that generates a better match of CryoSat-2 and SARAL data with ICESat-2 data, but it is not formally proved in this study that this correction allows for a more accurate roughness quantification. For this, roughness data at all the relevant scales (1cm – 100m) should be used, as obtained though terrestrial LiDAR, UAV photogrammetry , eddy covariance data or even manual probing.

**Author Response:** Thank you for the comment. This is a subtle point of our results that we as authors have continually struggled with tyring to communicate clearly. We do not contend that our results nor ICESat-2 ATL06 products can recover feature-scale surface roughness. In fact, the smallest baseline in the RMS deviation that is recovered from along-track ICESat-2 ATL06 data is on the order of tens of meters. We rely on ATL06 though because it is a contemporaneous, ice sheet-wide and standard data product that we can compare our results against. Future studies could go into the ATL03 (photo-cloud data) to get to higher resolutions, however this is outside the scope of the here presented study.

What we observe is that our radar-based roughness results tend to align with a piecewise linear portion of the RMS deviation profiles derived from laser altimetry (e.g., Figure 2c and 2d). This linear portion lies within a range of baselines that should be reliably recoverable from both the ALS and ATL06 datatsets. The empirical correction of Equation 4 is meant to more reliably recover that agreement (e.g., Figures 7 and 8).

What this means though is that the roughness results we derive through the RSR analysis are not the true roughness values at the radar wavelength baseline. They are the projection of the larger baseline scale dependent behaviour to the wavelength scale. Based on the ALS data (Figure 2), we know there is often an inflection point in the RMS deviation profiles near 100 m baselines, which leads to our 'projected' wavelength scale RSR roughness likely underestimating the true value. But the RSR results remain representative of roughness at the larger baselines.

We agree that this very subtle but important point doesn't come across as clearly as we would have liked in the manuscript. We will modify Section 6.1 to try and make it clearer and a pointer that other datasets will be needed to fill these gaps is included in the revisions made to the reviewers previous comment.

**Actions Taken:**

The following changes have been made in Section 6.1.

*Instead, for the Ku- and Ka-band airborne and satellite radar altimetry data, RSR surface roughness is best interpreted* **not as the true wavelength-scale RMS deviation, but the projection of the scale-dependent behaviour observed at baselines between hundreds of metres and a few kilometres to the wavelength scale** . *The implication is then that the SARAL and CryoSat-2 RSR surface roughness results only have physical meaning far beyond the individual roughness feature scales currently considered critical in heat flux and SMB* **studies**. *As such, they have no direct role to play in the improvement of current GrIS SMB modelling. The relevance only at long baselines is likely also the reason why the surface roughness timeseries presented in Figure 10 do not exhibit the strong seasonal variability that has been reported in derivations of aerodynamic roughness lengths (Smeets and van den Broeke, 2008; van Tiggelen et al., 2021, 2023).*

Furthermore, in Section 4 (L252-253), a main result of this work is given based on the match that is found between the RSR derived roughness from the airborne radars, and the RMS

roughness from ALS "at larger baselines (e.g. between 200 and 700 m)" (L249). Perhaps this is a lack of understanding of the RSR algorithm on my part, but I do not understand how the radar RSR roughness and the laser RMS roughness are related to each other. These two methods seem fundamentally very different and sensitive to different scales of surface roughness: ~radar wavelength scale (cm) for radar RSR, and >100 m for ALS and ICESat-2 RMS, depending on the (arbitrary?) choice of the extrapolation interval. Hence the found match between RSR and RMS "between 200 and 700 m" in Fig 2 could as well be a coincidence, or the result of the postprocessing, such as how the data was detrended (L154). At least it does not seem to be physically based. Perhaps the authors could explain this better and perform a sensitivity test by varying the extrapolation intervals (e.g. 50-500m, 200m-600m, 300-1000m, etc…) and detrending methods (linear, polynomial, high-pass filtering,...)

**Author Response:** Thank you for the comment. We will start with the second half of the reviewers comment first.

First, we want to centre our analysis with a common surface roughness metric. Detrending elevation measurements with a constant linear background slope is the standard first step when calculating the RMS deviation, such that the mean value is zero (Shepard et al., 2001). That is why we default to that option instead of other approaches (e.g., polynomial, high-pass filtering etc.). We will include a citation in Section 3.1 to support the use of a linear background removal.

Second, the 200-700 m interval is not arbitrary but relates directly to the observed piecewise linear behaviour of the RMS deviation profiles derived from ALS and ICESat-2 ATL06 data (Figures 2 and 4). This range encompasses where the RMS deviation profiles exhibit a constant slope. Changing the extrapolation intervals outside of this range means moving away from that observed and repeated linear behaviour, the projection of which roughly aligned with the initial interpretation of the CryoVEx RSR results (i.e., scale-independent RMS heights). We will revise Section 4.1 to strengthen the reasoning behind our choice of interval.

Turning to the first half of the reviewer's comment, we consider roughness as a spectrum, sensitive to the distance over which it is measured (hence our use of the RMS deviation) as opposed to single constant number. In this way, the radars and lasers sample the same spectrum but in different ways. Roughness for radars is typically expressed relative to the short wavelengths but the illuminated portion of the surface can be substantial (meters to kilometres depending on application and processing) and for RSR analysis we are pulling together radar echo powers that can be 10's of kilometres apart. The purpose of this study is to use the more generally accepted laser surface roughness to establish an objective version of that spectrum; thereby providing a framework for interpreting the radar results.

**Actions Taken:**

The following citation has been added in Section 3.1.

*Figure 1 presents a simplified overview of how to calculate the RMS deviation at various baselines from a set of measured surface elevations along a profile **following Shepard et al. (2001)**.*

The following revisions have been made in Section 4.1 to strengthen the reasoning behind our choice of extrapolation interval.

*The continuation of the piecewise linear trends in RMS deviation from baselines less than 50 m to the radar wavelengths would overestimate the RSR results by roughly two orders of magnitude. However, projecting **the linear** scale-dependent behaviour at **longer** baselines (e.g., between 200 and 700 m) yields the dashed lines in Figures 2c**b** and 2d**e**, which closely intersect with the **initial values** derived from the ASIRAS and KAREN. This suggests that **surface roughness** derived through the RSR analysis of CryoVEx radar altimetry surface echo powers **are not scale-independent RMS heights, but** the wavelength-scale projection of surface roughness behaviour observed at long baselines. It should be noted, however, that this interpretation assumes that there are no further inflexion points at baselines smaller than those that can be accessed via the ALS data.*

Finally, as a non-radar expert yet interested in improved SMB modelling, I am hoping that the estimated RSR roughness (with values between 0.01 mm and 1 mm on the ice sheet) could be translated to a more useful metric (such as standard deviation of heights, average obstacle height, etc... ), which is then directly compared and shown as a map together with the same metric as estimated from ICESat-2 in Figure 10. It becomes clear at the end in section 6.1 that this new map can't unfortunately be used directly for aerodynamic roughness calculations, and I agree, yet I would invite the authors to make this clearer from the start and in the abstract/conclusion, and propose possible alternatives to overcome this limitation. Could this not be expected before the analysis, given the large footprint of the satellite radars?

**Author Response:** Thank you for the comment. The purpose of this study is to explore whether this expectation (i.e., RSR roughness being relevant to determining aerodynamic roughness) is valid or not. If the 'objective' surface roughness had proven scale-independent (e.g., a flat RMS deviation profile) and could be quantitatively linked to the radar results, then the RSR results could have had direct meaning for SMB calculations. Alas this turned out not to be the case; however, we do not think we could have made this assertion before performing the exercise. We agree though that the outcomes of the study will have to be more clearly stated in the Abstract and will adjust it as such.

**Actions Taken:**

The following changes have been made to the Abstract to more clearly communicate the results of the study.

*Surface roughness is an important factor to consider when modelling mass fluxes at the Greenland Ice Sheet (GrIS) surface (i.e., surface mass balance, SMB). This is because it can have important implications for both sensible and latent heat fluxes between the atmosphere and the ice sheet and near-surface ventilation. While surface roughness can be quantified from ground-based, airborne and spaceborne observations, satellite radar datasets provide the unique combination of long-term, repeat observations across the entire GrIS and insensitivity to illumination conditions and cloud cover. In this study, we investigate the reliability and*

*interpretation of a new type of surface roughness estimate derived from the analysis of Ku- and Ka-band airborne and spaceborne radar altimetry surface echo powers by comparing them to contemporaneous laser altimetry measurements. Airborne data are those acquired during the 2017 and 2019 CryoVEx campaigns while the satellite data (ESA CryoSat-2, CNES/ISRO SARAL, and NASA ICESat-2) are those acquired in November 2018. Our results show* **GrIS surface roughness is typically scale dependent. A**  *revised empirical mapping* **between** *quantified radar backscattering* **and** *surface roughness gives a better match to the coincident laser altimetry observations than an analytical model that assumes scale-independent roughness.* **Surface roughness derived from the radar surface echo powers are best interpreted not as the wavelength-scale RMS deviation representative of individual features, but the continued projection of scale-dependent roughness behaviour observed at hundreds of meter baselines down to the radar wavelength. This implies that the relevance of these roughness estimates to current SMB modelling efforts is limited, as surface roughness is treated as a homogenous and scale independent parameter**~~We also show that the radar altimetry-derived surface roughness is best interpreted as the wavelength-baseline linear projection of the scale-dependent surface roughness observed at hundreds of meter scales and is therefore not representative of individual small-scale features. These results provide critical context for interpreting the datasets and evaluating their applicability in modelling GrIS SMB~~.

Despite the limitations, perhaps an assumption could be made that the RMS between a certain scale range (e.g. 1 cm – 100m) is relevant for surface drag calculations, which can be indirectly estimated from the RSR or RMS algorithms. A typical roughness height from CryoSat-2, SARAL and ICESat-2 for these scales could be plotted as the final result, thereby paving the way for improved SMB modelling.

**Author Response:** Thank you for the comment. While we'd really like to provide a metric that is directly relevant to SMB modelling efforts (it would be a highlight of the study), at this time we just do not think the available data supports it. The ATL06 ICESat-2 data and our understanding of our RSR results really don't really have much to say on surface roughness at scales less than 100 m, which is where a lot of models place their emphasis. A key takeaway from this work though is to highlight the scale-dependency in surface roughness that, to our knowledge, is currently not represented in SMB modelling at all. The preceding revisions in response to the reviewer comments have tried to accentuate this point.

**Actions Taken:**

Revisions made in response to previous reviewer comments have tried to highlight the scale dependency in surface roughness observed in the remote sensing datasets.

**Other comments**

Title: Consider removing "Insights into" .

**Actions Taken:** We have revised the title following the reviewer's suggestion.

* Greenland Ice Sheet Surface Roughness from Ku-/Ka-band Radar Altimetry Surface Echo Strengths*

L16: (mentioned in Major comments) As far as I understand, it is an assumption by the authors and not a proved result that the revised empirical correction gives a better match to ICESat-2 because the "surface roughness across the GrIS is primarily scale-dependent". The better match is obtained by construction.

**Author Response:** Thank you for the comment. We believe we have responded to this specific comment while considering the major comment above. And note that scale dependent surface roughness is not implied by the RSR results but by the RMS deviation profiles derived from ICESat-2. The better match in this specific case is shown in Figure 7 where the SPM consistently overestimates roughness while the empirical fit does not.

**Actions Taken:** Please see the revisions made in response to the previous major comment from the reviewer.

L20: Please rephrase "wavelength-baseline linear projection of the scale-dependent surface roughness".

**Author Response:** Thank you for the comment. We agree that was a very confusing sentence. We have since revised it when updating the Abstract in response to one of the reviewer's major comments.

**Actions Taken:** Please see the revisions made in response to the previous major comment from the reviewer.

L40: Rephrase into "Numerical estimates of GrIS SMB are based on coupled climate/subsurface models".

**Actions Taken:** The sentence has been revised as suggested by the reviewer.

*Numerical estimates of GrIS SMB are based on coupled climate/subsurface models (van den Broeke et al., 2023)*

L85: What are conventional surface roughness metrics ?

**Author Response:** Thank you for the comment and re-reading this sentence again, we find it misleading. We state metrics in plural but ultimately only one metric is investigated as part of the study. We will revise the sentence to be more direct. We hope this will also address the reviewers comment.

**Actions Taken:** The sentence has been revised as follows.

*In this study, the reliability of the Ku- and Ka-band surface roughness estimates derived from the RSR analysis of radar altimetry surface echoes is assessed by comparing them to* **RMS**

*deviations (Shepard et al., 2001)* *derived from laser altimetry measurements.*

Section 2.1: please include some information about the ALS data processing and accuracy (also mentioned in major comments)

**Author Response:** Thank you for the comment. We believe we have addressed the reviewer's concerns as part of our revisions to one of the reviewer's earlier major comments.

Figure 1: Is this actual data from the ALS ? If so, please include the location and time of this measurements. If not, consider using real data with a quantitative x-axis which would help the reader to better understand the "raw" data.

**Author Response:** Thank you for the comment and to be clear, these are not real data. We have not gone looking for any of the raw ALS data, only the final gridded data products. Figure 1 is purely for demonstrating how an RMS deviation profile is constructed in the abstract. But we agree that having a more realistic x axis could be beneficial and will adjust the figures as such.

**Actions Taken:** A revised Figure 1 has been included in the manuscript.

[Figure]

We have also revised the figure caption to make it explicit that these are fictional data and not representative of any of the laser altimetry data used in this study.

*Figure 1: Simplified diagram demonstrating how to derive an RMS deviation [v(Δx)] profile that quantifies surface roughness as a function of horizontal baseline (Δx)* ***for an artificial roughness dataset****. The profile comprises the RMS of height deviations [i.e., elevations minus a background plane; z(x_i )] considering all combinations of points with a similar horizontal baseline. The RMS deviation profile [b)] can be derived from profile data (i.e., as demonstrated here) as well as two-dimensional surface elevation datasets.*

L154: Wouldn't it be more consistent to filter out the same larger wavelengths using the same high-pass filter for all ALS and ICESat-2 data ? The background plane depends on the length of the dataset, which makes it hard to compare short ALS data (1 km) with longer ICESat-2 data (10km), as done later in Figure 4.

**Author Response:** Thank you for the comment. During the analysis, there was a discussion on how to best define the background plane for the various surface elevation datasets that we ultimately want to derive RMS deviations from. In the end, the decision to use individual dataset-specific background planes as opposed to, as the reviewer suggests, a common one is

because of how the RMS deviation is calculated. It expects the height deviations (i.e., surface elevations minus background plane) to be distributed around a mean of zero. As we can see in Figures 3c and 3d, if we had removed the ICESat-2 background plane from the ALS data, the ALS height deviations would not have met this criterion (the local areas stand consistently higher than the background plane). The end result would have been much larger ALS RMS deviations, and this would have obscured the fact that relative to the more regional context, the areas immediately surrounding T30 and T41 are actually quite smooth. Furthermore, the RSR roughness is sensitive to the local area only and artificially boosting ALS roughness by considering different regional background plane would complicate the comparison. Finally, we think that highlighting that surface roughness is not a static metric but that is has multiple scale-dependencies is valuable. We agree that our rationale for using dataset-specific background planes needs to be more explicit in the manuscript.

**Actions Taken:** The following revision has been made in Section 3.1.

*Figure 3 presents the local surface elevations (Figures 3a and 3b) along with the height deviations (elevations minus the constant location-specific background plane; Figures 3c and 3d) surrounding the T30 and T41 ALS datasets. Elevations are taken from the 10 m ArcticDEM mosaic (Porter et al., 2018) and the background plane is defined using all 10 m ArcticDEM data within 20 km of the CryoVEx in-situ measurement location. **Dataset-specific (i.e., ALS, ICESat-2, ArcticDEM) background planes are used to ensure all calculated height deviations have a mean of zero (Shepard et al., 2001) and accurately reflect the local conditions over which the data are acquired (needed when comparing to the RSR results).** It is clear that the topographic variability is very small at T30 and essentially non-existent at T41.*

L182: How was this 1.5 degree threshold chosen? What are the potential effects of off-nadir instrument point on the results?

**Author Response:** Thank you for the comment. The off-nadir pointing threshold was chosen empirically in order to identify and remove data when the aircraft is turning.

Off-nadir pointing affects the radar echo power reflected from the ice sheet surface. The reflecting behaviour of the GrIS appears to be specular (i.e., mirror like), so most normally-incident radar energy reflected from the surface comes back to the antenna. This contrasts with diffuse behaviour, where incident radar energy would be scattered in all directions. If the aircraft is turning and the antenna is pointed off-nadir, energy reflecting from a dominantly specular surface will not be backscattered towards the spacecraft but reflected away from it and the echo power will contain only the weaker, diffusely scattered component. So off-nadir pointing leads to reductions in surface echo powers that are not attributable to changes in GrIS surface conditions. That is why these measurements are identified and removed.

That is not to say that these data are not valuable from the altimetry perspective, just that the power in the returned echo will be modified by a rolling aircraft. We will revise the manuscript to make this more explicit.

**Actions Taken:** The following revisions have been made in Section 3.2.

*Once extracted, ASIRAS and KAREN surface power amplitudes are omitted from further analysis if the measured roll is greater than 1.5 degrees. **This threshold is determined empirically and used** to limit the potential effects  a rolling aircraft and off-nadir instrument pointing **have on the strength of the surface echoes**.*

L189: I would strongly recommend that the authors better explain how the coherent and incoherent powers are estimated, as these are the backbone of the methods. Perhaps some formulas or a schematic (possibly in the appendix) would be beneficial.

**Author Response:** Thank you for the comment. The development of the RSR approach and all its nuances are captured in a line of papers stretching back to Grima et al. (2012). As our goals lie in the application of the method, we believe that getting too much into the fine details of how it works would clutter the study and serve as a distraction. That being said, we appreciate that more details would help elucidate the technique for those who have not come across it before. We will include revisions to this effect.

**Actions Taken:** The following revisions have been made in Section 3.2.

*The set of radar surface echo amplitudes closest to some defined location is used to construct **a histogram** and  statistical descriptors of the homodyned K-distribution fit **(i.e., metrics quantifying the mean and spread)** define the coherent (Pc) and incoherent (Pn) powers (Grima et al., 2014a).*

L194-208. These sentences could be shortened while still conveying the same information and being clearer at the same time.

**Author Response:** Thank you for the comment. We will endeavour to make this section clearer and more concise.

**Actions Taken:** The following revisions have been made in Section 3.2.

*For both ASIRAS and KAREN, **RSR histograms are constructed from** the 1000 closest surface echo powers to each in-situ position (measured in EPSG:3413). For the satellite **datasets, the CryoSat-2 LRM and SARAL results are taken directly from Scanlan et al. (2023) (i.e., five-by-five kilometre grid, 1000 surface echo powers closest to each grid node).**  For the CryoSat-2 SARIn data, the same five-by-five kilometre **grid is** used, but instead of 1000 **surface echo powers**,  the 12,000 closest samples **are used** to overcome the apparent statistical dependence of adjacent CryoSat-2 SARIn surface echo powers noted in Scanlan et al. (2023).*

*Two metrics are used to **quality control**  the RSR results: first, the **distance to the furthest surface echo power measurement considered**, and second, the correlation coefficient between the observed surface echo power histogram and the*

*statistical̶h̶o̶m̶o̶d̶y̶n̶e̶d̶ ̶K̶ ̶d̶i̶s̶t̶r̶i̶b̶u̶t̶i̶o̶n̶ fit. The former is irrelevant for the CryoVEx data as we mandate using the closest 1000 data points surrounding the in-situ locations regardless of how far they may be. For the satellite results, **search radii of 50 km, 25 km, and 40 km are used for the CryoSat-2 LRM, CryoSat-2 SARIn, and SARAL results** ̶r̶e̶s̶p̶e̶c̶t̶i̶v̶e̶l̶y̶ ̶u̶s̶e̶ ̶t̶h̶e̶ ̶f̶o̶l̶l̶o̶w̶i̶n̶g̶ ̶m̶a̶x̶i̶m̶u̶m̶ ̶s̶e̶a̶r̶c̶h̶ ̶r̶a̶d̶i̶i̶:̶ ̶5̶0̶ ̶k̶m̶ ̶f̶o̶r̶ ̶C̶r̶y̶o̶S̶a̶t̶-̶2̶ ̶L̶R̶M̶,̶ ̶2̶5̶ ̶k̶m̶ ̶f̶r̶o̶m̶ ̶C̶r̶y̶o̶S̶a̶t̶-̶2̶ ̶S̶A̶R̶I̶n̶,̶ ̶a̶n̶d̶ ̶4̶0̶ ̶k̶m̶ ̶f̶o̶r̶ ̶S̶A̶R̶A̶L̶ (Table 1). The CryoSat-2 LRM and SARAL maximum search radii are the same as those used in Scanlan et al., (2023), while the CryoSat-2 SARIn radius **is extended as more datapoints are being**̶h̶a̶s̶ ̶b̶e̶e̶n̶ ̶e̶x̶t̶e̶n̶d̶e̶d̶ ̶t̶o̶ ̶a̶c̶c̶o̶u̶n̶t̶ ̶f̶o̶r̶ ̶t̶h̶e̶ ̶i̶n̶c̶r̶e̶a̶s̶e̶d̶ ̶n̶u̶m̶b̶e̶r̶ ̶o̶f̶ ̶e̶c̶h̶o̶ ̶s̶a̶m̶p̶l̶e̶s̶ considered. **A minimum correlation coefficient of 0.96 is required for all RSR results**̶F̶o̶r̶ ̶a̶l̶l̶ ̶r̶a̶d̶a̶r̶ ̶a̶l̶t̶i̶m̶e̶t̶r̶y̶ ̶d̶a̶t̶a̶s̶e̶t̶s̶,̶ ̶t̶h̶e̶ ̶c̶o̶r̶r̶e̶l̶a̶t̶i̶o̶n̶ ̶c̶o̶e̶f̶f̶i̶c̶i̶e̶n̶t̶ ̶m̶u̶s̶t̶ ̶e̶x̶c̶e̶e̶d̶ ̶0̶.̶9̶6̶ ̶f̶o̶r̶ ̶t̶h̶e̶ ̶R̶S̶R̶ ̶m̶e̶a̶s̶u̶r̶e̶m̶e̶n̶t̶ ̶t̶o̶ ̶b̶e̶ ̶c̶o̶n̶s̶i̶d̶e̶r̶e̶d̶ ̶v̶a̶l̶i̶d̶ (Grima et al., 2012, 2014b, a; Scanlan et al., 2023).*

L200: what data selection / what metrics have been applied to ensure quality of the ICESat-2 data ?

**Author Response:** Thank you for the comment. We agree that some of the details regarding the handling of the ICESat-2 data were mistakenly omitted from the original manuscript. We will revise this in our updated version.

**Actions Taken:** The following revisions have been made in Section 2.2.

*Leveraging the results of Scanlan et al. (2023), because CryoSat-2 operates in two different acquisition modes while over Greenland, this study makes use of both Low-Resolution Mode (LRM) Level 1B (L1B) and SAR Interferometric (SARIn) Full-Bit Rate (FBR) CryoSat-2 Baseline-D data products. The LRM data products cover the GrIS interior, where CryoSat-2 waveforms are stacked from an individual 1.97 kHz pulse repetition frequency (PRF) to a constant data rate of 20 Hz. Across the GrIS margins, the CryoSat-2 SARIn data products contain reflected waveforms from 18.181 kHz PRF, 64-pulse bursts (burst repetition frequency of 21 Hz) acquired by both CryoSat-2 receive antennas. The FBR data are preferred to higher level SARIn data products (e.g., Level 1) as they have not yet been subject to any SAR focusing. The SARAL Sensor Geophysical Data Record (SGDR) data products are similar to CryoSat-2 LRM products in that initially 3.8 kHz PRF waveforms are averaged alongtrack in 25 ms (40 Hz) intervals. Finally, this study makes use of **release 005** ATL06 ICESat-2 data products providing geo-located surface heights across the GrIS in 20-meter increments. **The quality summary variable included with each of the six beams in the ATL06 data products is used to reject lower certainty surface height measurements.***

L209-211: Is it possible that some words are missing from this sentence ?

**Author Response:** Thank you for the comment. We don't think there are words missing from this particular sentence, but we agree the thoughts behind it could be stated more clearly.

**Actions Taken:** The following revisions have been made in Section 3.2

***Once coherent (Pc) and incoherent (Pn) powers are determined***̶F̶r̶o̶m̶ ̶t̶h̶e̶ ̶q̶u̶a̶l̶i̶t̶y̶ ̶c̶o̶n̶t̶r̶o̶l̶l̶e̶d̶ ̶R̶S̶R̶ ̶r̶e̶s̶u̶l̶t̶s̶*, **deriving** surface roughness **requires** ̶i̶s̶ ̶d̶e̶r̶i̶v̶e̶d̶ ̶b̶y̶ adopting a representative backscattering model. The simplest implementation of the RSR technique (Grima et al., 2012,*

*2014b, a; Scanlan et al., 2023) assumes incoherent backscattering from the surface follows the Small Perturbation Model (SPM) (Ulaby et al., 1982).* **In this case,**  *surface RMS height can be derived*  *following*

L214: What is effectively the highest sigma_h that can be estimated using CryoSat-2 using this method ? How does this compare to the RMS of the actual expected topography on the Greenland ice sheet ? If this method only works for sigma_h < ~6 mm (for CryoSat-2), that leaves out most of the ice sheet surface with ice hummocks, sastrugi, dunes, etc…

**Author Response:** Thank you for the comment. The 6 mm number comes from the demonstrated validity of the Small Perturbation Model (SPM) describing radar backscattering from a slightly rough surface, not the maximum detectable roughness from RSR. RSR outputs coherent and incoherent powers only and it is through the backscattering model that these are inverted to roughness. A different backscattering model (e.g., Kirchoff, Integral Equation Method) would have a different range of validity and many extend to larger RMS height values. The drawback though of many of these other models through is that they have many more degrees of freedom, which complicates unique interpretations of the RSR results. The purpose of this study is to put the assumption of the SPM to the test against surface roughness derived from laser altimetry. And from Figure 7, we can see the SPM is clearly lacking and tends to actually overestimate roughness. We also find that the RSR-derived roughness is not directly relevant at the scales of hummocks and sastrugi.

**Actions Taken:** We have chosen not to make any revisions in the manuscript in response to this specific point as we feel we have made progress clarifying how to interpret the RSR roughness results in the context of other revisions and, specifically, in a revised Section 6.1.

L215: please define the "surface correlation length".

**Author Response:** Thank you for the comment. The correlation length is a roughness-related parameter in the SPM that describes a typical length-scale over which the roughness occurs. In an analysis of the RSR results using the SPM, it is often assumed that this length-scale is greater than the radar footprint, which simplifies the underlying equations (see Grima et al., 2014a https://doi.org/10.1016/j.pss.2014.07.018). We agree that this should be mentioned more clearly in the manuscript and will make a corresponding revision

**Actions Taken:** The following revision has been made in Section 3.2.

*The analytical relationship for deriving RMS height from the RSR results in Equation 3 is based on the assumption that the surface roughness correlation length* **(i.e., the length scale over which the roughness occurs)** *is* **large relative to the radar footprint and can**  *be neglected (Grima et al., 2012, 2014a).*

L230: How are the densities recovered using RSR ? Consider adding a bit more background information in Section 3.2.

**Author Response:** Thank you for the comment. We really want to keep the focus of the manuscript on the roughness results. That is why we do not spend much time discussing the derivation of density and only present it as a method applicability check in Figure 2. However,

the reviewer's point is well-taken and we can see the value of including a bit more background in Section 3.2. We will include the equation that relates Pc to the Fresnel reflection coefficient and close the circle for deriving surface properties from RSR results.

**Actions Taken:** The following has been added at the end of Section 3.2.

*Once calibrated, the coherent powers returned from the RSR analysis are related to the Fresnel reflection coefficient for an air-snow/firn interface ($r = (1 - \sqrt{\varepsilon_r})/(1 + \sqrt{\varepsilon_r})$ where $\varepsilon_r$ is the relative dielectric permittivity of the snow/firn) following*

$$P_{c,cal} = r^2 e^{-(2k\sigma_h)^2} . \qquad (4)$$

*Multiple empirical functions exist for then converting relative dielectric permittivity to snow density (Ambach and Denoth, 1980; Kovacs et al., 1995; Pomerleau et al., 2020; Tiuri et al., 1984).*

Figure 2: In panels c and d, the triangles can't be discerned. In panel d, the resolution of the T21 ALS seems much higher than for the other locations.

**Author Response:** Thank you for the comment. Regarding the individual data points, we've revised Figure 2 with a zoomed in window to try and show them better.

As for the T21 resolution, it is higher than the other two locations. This is why there is one point at a baseline less than 1 m. The CryoVEx aircraft flew over T9 and T12 only once during data collection, so the ALS data exist only in the processed 1-by-1 m grid. In contrast, the aircraft overflew T21 multiple times in different azimuthal directions. The effect is then that when data from the multiple groundtrack segments are analysed together, where the groundtracks overlap, the spacing of the combined surface elevation dataset can be finer than the one metre spacing. This is why it is possible to look at smaller baselines at T21 compared to T9 and T12. To emphasize this point the following figures present the processed 2019 ALS data within 1 km of T9, T12, and T21.

[Figure]

**Actions Taken:** The revised Figure 2 has been included in the manuscript

[Figure]

The following revisions have been made to the Figure 2 caption.

*Figure 2: Results from the analysis of 2017 and 2019 airborne CryoVEx data. The locations of the in-situ measurements along the EGIG line are presented in Panel a). Panel b) presents the agreement (R of 0.45) between the RSR-derived densities from ASIRAS (triangles) and KAREN (square) surface echoes powers with those measured in-situ. The comparison of RMS heights from the radar altimetry (assumed to be equivalent to the RMS deviation at the radar wavelength baseline) and the ALS data (circles) RMS deviations as a function of baseline are presented in Panels c) (CryoVEx 2017) and d) (CryoVEx 2019).* **Inserts in panels c) and d) presented zoomed in views of the RSR results.** *The RSR-based roughness estimates align well with the projection of the piecewise linear portion of the ALS RMS deviations profiles between 200-700 m (i.e., the grey region) to the radar wavelength baseline.*

L239 On what is the assumption based that the RSR radar roughness corresponds to the RMS at the wavelength scale? please also specify "radar wavelength" to avoid confusion about which wavelength was used. How were the shaded areas 200-700 m chosen if not arbitrarily?

**Author Response:** Thank you for the comment. Regarding the latter portion of the reviewer's comment, please see our response to the reviewer's major comment above. Essentially, the 200-700 m interval is not arbitrary but aligns with a consistent piecewise linear portion of the RMS deviation profile detectable in both ICESat-2 and ALS data. Using other intervals would mean combining seemingly different roughness regimes (i.e., piecewise linear portions). In terms of the former, the relation between backscattered radar powers and roughness at the

wavelength scale is a commonly invoked relationship. We agree that specificity was missing when discussing the wavelength, and we will make revisions to make this point clearer.

**Actions Taken:** The following revision has been made to the Figure 2 caption.

*Figure 2: Results from the analysis of 2017 and 2019 airborne CryoVEx data. The locations of the in-situ measurements along the EGIG line are presented in Panel a). Panel b) presents the agreement (R of 0.45) between the RSR-derived densities from ASIRAS (triangles) and KAREN (square) surface echoes powers with those measured in-situ. The comparison of RMS heights from the radar altimetry (assumed to be equivalent to the RMS deviation at the **radar wavelength baseline**) and the ALS data (circles) RMS deviations as a function of baseline are presented in Panels c) (CryoVEx 2017) and d) (CryoVEx 2019). Inserts in panels c) and d) presented zoomed in views of the RSR results. The RSR-based roughness estimates align well with the projection of the piecewise linear portion of the ALS RMS deviations profiles between 200-700 m (i.e., the grey region) to the radar wavelength baseline.*

L244: It took me quite some time to understand Figure 2c/d, so perhaps "immediately clear" is not the most accurate phrasing

**Author Response:** Thank you for the comment. The reviewer's point is well-taken we will adjust the manuscript to make it clearer what is being referred to.

**Actions Taken:** The following revisions have been made in Section 4.1

***The ALS-derived RMS deviation profiles presented in Figures 2c and 2d exhibit surface roughness behaviour that does not immediately align with the RSR results. At short (<100 m) baselines, the ALS RMS deviation profiles are relatively flat and therefore not strongly dependent on the horizontal scale over which roughness is measured. The situation changes though at long baselines (>100 m) where there is a stronger scale-dependency as progressively larger RMS deviations are measured for longer baselines. The fundamental pattern is then one of two piecewise linear components; a flatter, less scale-dependent behaviour at small baselines, and more scale-dependency at longer baselines.*** ~~*It is immediately clear from Figure 2 that the ASIRAS and KAREN surface roughness estimates do not agree with a direct continuation of the shortest baseline scale laser surface roughness behaviours down to the wavelength scale. While RMS deviation typically exhibits a strong scale dependency for baselines greater than 100 m (i.e., RMS deviation increases with baseline), for shorter baselines, the RMS deviation profiles level off, revealing more scale-independent behaviour.**The*~~ *continuation of the piecewise linear trends in RMS deviation from baselines less than 50 m to the radar wavelengths would overestimate the RSR results by roughly two orders of magnitude.*

L252-253: mentioned in major comments

**Author Response:** Thank you for the comment. Please see our response and actions above.

Figure 3: Please adjust the color scale of Fig3d.

**Author Response:** Thank you for the comment. The immediate area surrounding T41 stands roughly 5 m higher than the background plane and is very flat. The original colorbar limits were chosen to emphasize this point (all the same shade of red and no blue). But we agree small height deviations are difficult to see. We will remake Figure 3 but will use a different colorbar since the there are no negative height deviations to plot. We believe that using the same diverging colorbar as Figure 3c would be misleading.

**Actions Taken:** The following updated version of Figure 3 has been included in the revised manuscript.

[Figure]

We have also revised the Figure 3 caption to point out the difference between the Figure 3c and 3d colorbars.

*Figure 3: Surface elevations [a) and b)] and height deviations [c) and d)] surrounding the T30 [a) and c)] and T41 [b) and d)] CryoVEx locations. The ALS data considered in deriving the corresponding RMS deviation profiles in Figure 2c come from within the black polygons. It is clear that T30 and T41 are sited in locally smooth regions of the GrIS.* **Note the change in colorbar and associated limits between c) and d).**

Figure 4: The ALS and ICESat-2 ATL06 data are very different and show different behavior in the 100m-1km range, which is unexpected. Is this due to the different detrending of the data ?

**Author Response:** Thank you for the comment. We don't think this behaviour is too unexpected; or at least it is explainable. Both datasets centre on the in situ locations but the ALS data are drawn from only the local area within 1 km of the T30 and T41 sites, while the ICESat-2 data are more regional due to a poorer spatial coverage. From Figure 3, we can see that the areas covered by the ALS data (i.e., within the black outlines) are much smoother than the regional conditions. This is why there is such a marked difference between the ALS and ICESat-2 RMS deviation profiles in the overlapping baseline range. ALS samples a much smoother local area within the rougher regional background ICESat-2 sees. We will revise the manuscript to make this point clearer.

**Actions Taken:** The following revisions have been made to the end of Section 4.1.

*To further emphasise the smoothness of the GrIS near T30 and T41, Figure 4 compares the ALS RMS deviation profiles with those derived from all ICESat-2 surface elevations within 25 km and 35 km of T30 and T41, respectively. We must use ICESat-2 data that are further away from the T30 and T41 sites because these locations are between ICESat-2 orbital ground tracks. When considering  surface topography over a broader **regional** area (i.e., ICESat-2), **the stronger scale dependency**  in surface roughness  is once again observed. **Less scale dependency exists in the ALS RMS deviation profiles between 100 m and 1 km because T30 and T41 are situated in a locally very smooth portion of the GrIS (Figure 3).***

L380: This is very interesting, but perhaps adding the elevation lines in Figure 9 would help to better place this specific area with anomalous RSR retrievals in a broader context. What are the ICESat-2 RMS values in this region ? Can this also be related to noise in the ICESat-2 data ?

**Author Response:** Thank you for the comment. We also agree that this result is very interesting and surprising. We were not expecting to see such a strongly localized anomaly in the RSR results so high on the ice sheet. To avoid cluttering the figure, instead of adding contour lines, we suggest including a note as to the elevation of the anomalous region in the Figure 9 caption.

The extrapolated RMS deviation values for the region are predominantly those in Figure 8a that lie outside the upper 2-sigma outlier cut-off (i.e., those in the top-left of the figure). We do not initially believe this is an issue originating with the ICESat-2 measurements because it does not appear aligned with orbital groundtracks and we reject low quality ICESat-2 surface elevations during the data ingestion phase.

**Actions Taken:** The following revision has been made to the Figure 9 caption.

*Figure 9: Locations of >2σ outliers between the wavelength-scale RMS deviations projected from ICESat-2 and a) CryoSat-2 and b) SARAL surface roughness estimates. The locations are plotted on top of maps showing the number of months with valid (i.e., quality-controlled) RSR observations for the period 2013-2018 (72 months). While some outlying roughness mismatches occur closer to the boundaries of the various datasets, there is a cluster in SE Greenland **at ~3000 m elevation in the vicinity of the ice divide** that corresponds with a zone of RSR results that do not meet the quality control criteria. The impact of CryoSat-2 and SARAL*

*orbital designs can be seen in the spatial patterns (CryoSat-2 SARIn latitudinal stripping and SARAL hatching) in the southern portions of the ice sheet.*

We have also expanded on how the quality of the input ICESat-2 data is controlled in response to one of the reviewer's earlier comments.

Figure 10: why is the average roughness only shown for 2 transects ? If the data is available, consider plotting the data for the entire ice sheet. Also, why is only SARAL roughness shown and not also CryoSat-2 and ICESat-2 ?

**Author Response:** Thank you for the comment. The reviewer makes a fair point. The ArcticDEM was originally chosen to accentuate major drainage basins for those who might be unfamiliar with the GrIS. We will follow their suggestion and revise the basemap to be the long-term (2013-2018) mean surface roughness from SARAL.

As to why SARAL data are plotted and not CryoSat-2, it is because the two datasets are scaled versions of each other. When the RSR derived surface roughness is normalized by the radar wavelength (e.g., Figures 6, 7, and 8), the CryoSat-2 and SARAL data lie directly on top of another. The roughness maps and timeseries have different absolute values but similar overall distributions. So we believe it is redundant to present both of them.

**Actions Taken:** The following revised version of Figure 10 has been included in the manuscript.

[Figure]

The Figure 10 caption has been modified as follows.

*Figure 10: 2013-2018 SARAL surface roughness mean [a)] and timeseries [b) and c)] along east-west and north-south transects cross-cutting the GrIS. While there is strong spatial variability in RSR-derived surface roughness across the six-year period (i.e., margins are rougher than the interior), the temporal variability in surface roughness is minor. *

The following has also been added in Section 5.2.

*The spatial patterns in surface roughness are similar to those presented in Scanlan et al. (2023) and highlight the expected pattern of a smooth ice sheet interior that becomes progressively rougher towards the margin and in the South.* ***Elevated surface roughness in the vicinity of the fast-flowing Northeast Greenland Ice Stream (NEGIS) is also clearly discernible.*** *The region of elevated surface roughness further inland from the margin at latitudes slightly less than 70° overlaps with the catchment of Sermeq Kujalleq (Jakobshavn Isbræ).*

L420: Consider renaming section 6.1

**Author Response:** Thank you for the comment. We agree that this probably wasn't the best choice of names for this section. We will revise the name to be more descriptive of the discussion that follows

**Actions Taken:** Section 6.1 has been renamed as follows

*6.1* ***Implications for SMB and Heat Flux Modelling***

L429: This seems to be the first moment where the actual topography of the ice sheet is discussed. Perhaps the typical surface features could be introduced in an earlier stage for better overall understanding of what the satellite radars are supposed to detect (e.g. Between sections 2 and 3, or in the introduction). Some inspiration for this purpose :

> Filhol, S., & Sturm, M. (2015). Snow bedforms: A review, new data, and a formation mode. *Journal of Geophysical Research: Earth Surface*, 1645–1669. https://doi.org/10.1002/2015JF003529

> Picard, G., Arnaud, L., Caneill, R., Lefebvre, E., & Lamare, M. (2019). Observation of the process of snow accumulation on the Antarctic Plateau by time lapse laser scanning. *Cryosphere*, *13*(7), 1983–1999. https://doi.org/10.5194/tc-13-1983-2019

> Zuhr, A. M., Münch, T., Steen-Larsen, H. C., Hörhold, M., & Laepple, T. (2021). Local-scale deposition of surface snow on the Greenland ice sheet. *The Cryosphere*, *15*(10), 4873–4900. https://doi.org/10.5194/tc-15-4873-2021

**Author Response:** Thank you for the comment. By no means were we intending to neglect or minimize previous studies of snow bedforms and their evolution. One of our aims in the Introduction is to establish the intersection between surface roughness and SMB/heat flux modelling regardless of the type of landform associated with that roughness and highlight the different ways roughness has been measured in practice. It is then in Section 6.1 where we take a closer look at the scales of roughness the modelling community considers the most important to SMB and heat flux studies and how that aligns with the more holistic treatment of roughness we've tried to maintain so far. These additional references are very useful though for further establishing the state-of-the-art in measuring ice sheet surface roughness and we will include them in the Introduction.

**Actions Taken:** The following revisions have been made in the Introduction.

*To extend the meter-scale local comb gauge/snow blade measurements, Herzfeld et al. (2000) developed a towed sensor capable of measuring surface elevations at fine spatial scales along profiles hundreds of meters in length.* ***Fixed, ground-based laser scanning measurements have also been used to characterize the two-dimensional distribution and growth of meter-scale snow bedforms (e.g., meter-scale dunes, sastrugi, etc.) (Filhol and Sturm, 2015; Picard et al., 2019; Zuhr et al., 2021).*** *While ground-based surface roughness measurements yield the finest spatial sampling, their large-scale applicability is limited as they are very time-consuming and subject to site accessibility (e.g., remoteness, weather, etc.) issues.*

L465: Please include in 3.2 how the density is computed, which helps the reader to understand this adjustment term.

**Author Response:** Thank you for the comment. We agree that this point is worthy of further emphasis in Section 3.2.

**Actions Taken:** We have included our suggested revision addressing this point in response to one of the reviewer's earlier comments.